# Drivers and $CO_2$ flux budgets in a Sahelian *Faidherbia albida* agro-silvo-pastoral parkland: Insights from continuous high-frequency soil chamber measurements and Eddy Covariance.

Seydina Mohamad Ba [a d], Olivier Roupsard [b c d], Lydie Chapuis-Lardy [c f], Frédéric Bouvery [g], Yélognissè Agbohessou [c e], Maxime Duthoit [c e], Aleksander Wieckowski [h], Torbern Tagesson [h], Mohamed Habibou Assouma [i j k], Espoir K. Gaglo [a d], Claire Delon [l], Bienvenu Sambou [a], Dominique Serça [l]

[a] Faculté des Sciences et Techniques (FST), Institut des Sciences de l'Environnement (ISE), Université Cheikh Anta Diop (UCAD) de Dakar, 5005, Dakar-Fann, Sénégal
[b] CIRAD, UMR Eco&Sols, Dakar, Sénégal
[c] Eco&Sols, Univ Montpellier, CIRAD, INRAE, Institut Agro, IRD, Montpellier, France
[d] LMI IESOL, Centre IRD-ISRA de Bel Air, Route des hydrocarbures, 18524, Dakar, Sénégal
[e] CIRAD, UMR Eco&Sols, Université de Montpellier, Cirad, INRAE, IRD, Institut Agro Montpellier, 2 place Viala, Montpellier, France
[f] IRD, UMR Eco&Sols, Université de Montpellier, Cirad, INRAE, IRD, Institut Agro Montpellier, 2 place Viala, Montpellier, France
[g] INRAE, 147 rue de l'Université, 75338 Paris, France
[h] Department of Physical Geography and Ecosystem Science, Lund University, Sölvegatan 12, S-223 62 Lund, Sweden
[i] CIRAD, UMR SELMET, dP ASAP, Bobo Dioulasso, Burkina Faso
[j] SELMET, CIRAD, INRAE, Univ Montpellier, Institut SupAgro, Montpellier, France
[k] Centre International de Recherche-Développement sur l'Élevage en zone Subhumide (CIRDES), N°559, rue 5-31 Avenue du Gouverneur Louveau, Bobo-Dioulasso, Burkina Faso
[l] Laboratoire d'Aérologie, Université de Toulouse, CNRS, IRD, 14 Avenue Edouard Belin, 31400 Toulouse, France

**Corresponding authors:**
Seydina Mohamad Ba: seydina.ba@ird.fr
Olivier Roupsard: olivier.roupsard@cirad.fr

**Highlights:**

- Long-term high frequency $CO_2$ flux measurements using automated static chambers in a Sahelian *F. albida* parkland.
- Empirical gap-filling and flux partitioning methods validated against Eddy Covariance GPP.
- Fluxes peaked during the rainy season in both FS and Sh, driven mainly by soil moisture and leaf area.
- *F. albida* trees enhance $CO_2$ fluxes under canopies ("fertile island" effect) and account for ~50% of annual ecosystem GPP.

**ABSTRACT**:

Agroforestry systems — combining trees with crops and/or livestock — are increasingly promoted as sustainable and climate-resilient land-use strategies. Despite their widespread presence in the Sahel, experimental data on their potential as carbon sinks are scarce. This study presents a full-year, high-frequency dataset of $CO_2$ fluxes in a Sahelian agro-silvo-pastoral parkland dominated by *F. albida*, located in Senegal's groundnut basin. $CO_2$ fluxes were continuously measured using automated static chambers, allowing the quantification of soil and crop respiration (Rch), gross primary production (GPPch), and net carbon exchange ($FCO_2ch$) under both full sun and shaded (under tree canopies) environments.

Seasonal patterns of $CO_2$ fluxes were similar in both environments, with peaks during the rainy season. Rch and GPPch were significantly higher under tree canopies, indicating a 'fertile island' effect. $CO_2$ flux variability was primarily driven by soil moisture and leaf area index. Chamber-based GPP estimates closely matched those from Eddy Covariance measurements. On an annual scale, *F. albida* trees contributed approximately 50% of total ecosystem GPP, with a carbon use efficiency of 0.48. Net annual $CO_2$ exchange was estimated at $-1.4 \pm 0.02$ and $-1.8 \pm 0.01$ Mg C-$CO_2$ $ha^{-1}$ using chamber and Eddy Covariance methods, respectively. These findings underscore the role of *F. albida*-based agroforestry systems as effective carbon sinks in Sahelian landscapes, supporting their potential contribution to climate change mitigation.

**Keywords:** Sahelian agro-silvo-pastoral systems, $CO_2$ fluxes, automated static chambers, Eddy Covariance, 'fertile island effect' of trees, carbon budgets.

## 1. Introduction

Plant photosynthesis and respiration —both autotrophic (plant) and heterotrophic (microbial)— are fundamental processes driving carbon dioxide ($CO_2$) fluxes in terrestrial ecosystems (Lambers et al., 2008; Raich et al., 2014; Reichle, 2020). Accurate quantification of these processes is critical for assessing ecosystem carbon (C) sink potential (Baldocchi, 2020), particularly for informing climate-smart land management strategies.

To capture these processes at the ecosystem scale, the Eddy Covariance (EC) technique has emerged as a transformative method, enabling continuous and high-frequency $CO_2$ flux measurements (Baldocchi, 2003, 2008). Extensive EC networks in Europe (Stojanović et al., 2024), Asia (Yu et al., 2011), and the Americas (Chu et al., 2021) have significantly advanced our understanding of the global C cycle. In contrast, sub-Saharan Africa remains critically underrepresented (Bombelli et al., 2009; Houghton & Hackler, 2006; Williams et al., 2007). Although some studies have used EC (Ardö et al., 2008; Brümmer et al., 2008; Merbold et al., 2009; Tagesson et al., 2016), static chambers (Assouma et al., 2017; Owusu et al., 2024; Rosenstock et al., 2016; Wachiye et al., 2020), or modeling approaches (Agbohessou et al., 2023, 2024; Delon et al., 2019; Rahimi et al., 2021), they remain sparse and methodologically heterogeneous, limiting comparability and regional C budget integration.

Among these underrepresented landscapes, agroforestry systems in the Sahel— particularly agro-silvo-pastoral systems (ASPS) that combine trees, crops, and livestock— are increasingly promoted for sustainable land management and climate resilience (Cardinael et al., 2021; Gupta et al., 2023; Mbow et al., 2014; Stetter & Sauer, 2024). However, the structural and functional heterogeneity of these systems poses significant challenges for accurately quantifying and upscaling C fluxes. *Faidherbia albida*, a keystone agroforestry tree species in these ASPS (Leroux et al., 2022; Lu et al., 2022), is of particular interest due to its reverse phenology, capacity to enhance soil fertility and crop yields (Bayala et al., 2020; Roupsard et al., 2020; Sileshi et al., 2016; 2020). Yet, its functional role in modulating both the magnitude and seasonal dynamics of $CO_2$ fluxes remains poorly understood.

Addressing this knowledge gap requires integrated approaches capable of capturing both aggregate and component-specific $CO_2$ fluxes. While EC remains the gold standard method for $CO_2$ flux measurements at the landscape scale (Baldocchi, 2003), it captures net ecosystem exchange (NEE) as an aggregate signal, without separating the contributions from individual compartments such as soil, crops, and trees. This limits its utility for disentangling processes and attributing sources in heterogeneous systems like ASPS. Automatic static chambers provide a valuable complement to EC, as they enable continuous, high-frequency measurements at finer scales and at the level of specific ecosystem components. This approach facilitates component-specific quantification of $CO_2$ fluxes, particularly from soil and crop compartments (Luo & Zhou, 2006;

Denmead, 2008; Zaman et al., 2021). When combined with EC, this dual-method approach strengthens source attribution and improves the upscaling of fluxes across complex agroforestry landscapes.

This study presents one of the first integrated quantification of $CO_2$ fluxes in a Sahelian ASPS dominated by *F. albida*, combining EC and automatic static chambers.

Specifically, we aim to (1) conduct year-round, high-frequency *in situ* $CO_2$ flux measurements from soil and crops using automated static chambers; (2) partition the net $CO_2$ fluxes ($FCO_2ch$) into respiration (Rch) and photosynthesis (GPPch); (3) investigate the environmental drivers of fluxes and the spatial variability linked to tree presence; and (4) compare chamber-based flux estimates with ecosystem-scale measurements derived from the EC method.

## 2. Materials and methods

### *2.1. Site description*

The study was conducted in the agroforestry parkland of Sob village (Niakhar municipality, Fatick region), located in the groundnut basin of Senegal, within the Sahelo-Sudanian climatic zone of West Africa (Fig. 1). The climate is characterized by a long dry season (8–9 months) with high temperatures and strong diurnal variations, and a short rainy season from late June to mid-October (Delaunay et al., 2018).

Soils are locally known as "*Dior"* and classified as Arenosols (IUSS Working Group WRB, 2022). The topsoil has low organic matter (<1%) and phosphorus (<3 mg $kg^{-1}$), a sandy texture (>85% sand), and an acidic pH (Malou et al., 2021; Siegwart et al., 2022). Rainfed agriculture predominates. The main cropping system includes pearl millet (*Pennisetum glaucum L.*) and groundnut (*Arachis hypogaea L.*) in biennial rotation, with occasional intercropping of cowpea (*Vigna unguiculata L.*).

The site hosts the 'Faidherbia Flux' station (14°29′44.916″N; 16°27′12.851″W; FLUXNET ID: SN-Nkr), a long-term research platform for monitoring ecosystem services in agroforestry systems. It is dominated by *F. albida*, a nitrogen-fixing, reverse-phenology tree with deep roots accessing groundwater (Roupsard et al., 1999). The tree density is ~13 trees $ha^{-1}$, with canopies covering ~10% of the soil surface (Roupsard et al., 2020). The EC tower is installed at 20 m height, approximately 12.5 m above the canopy. The study field is a typical 'bush field', characterized by low soil fertility, no mineral fertilization, and off-site export of crop residues and manure (Malou et al., 2021).

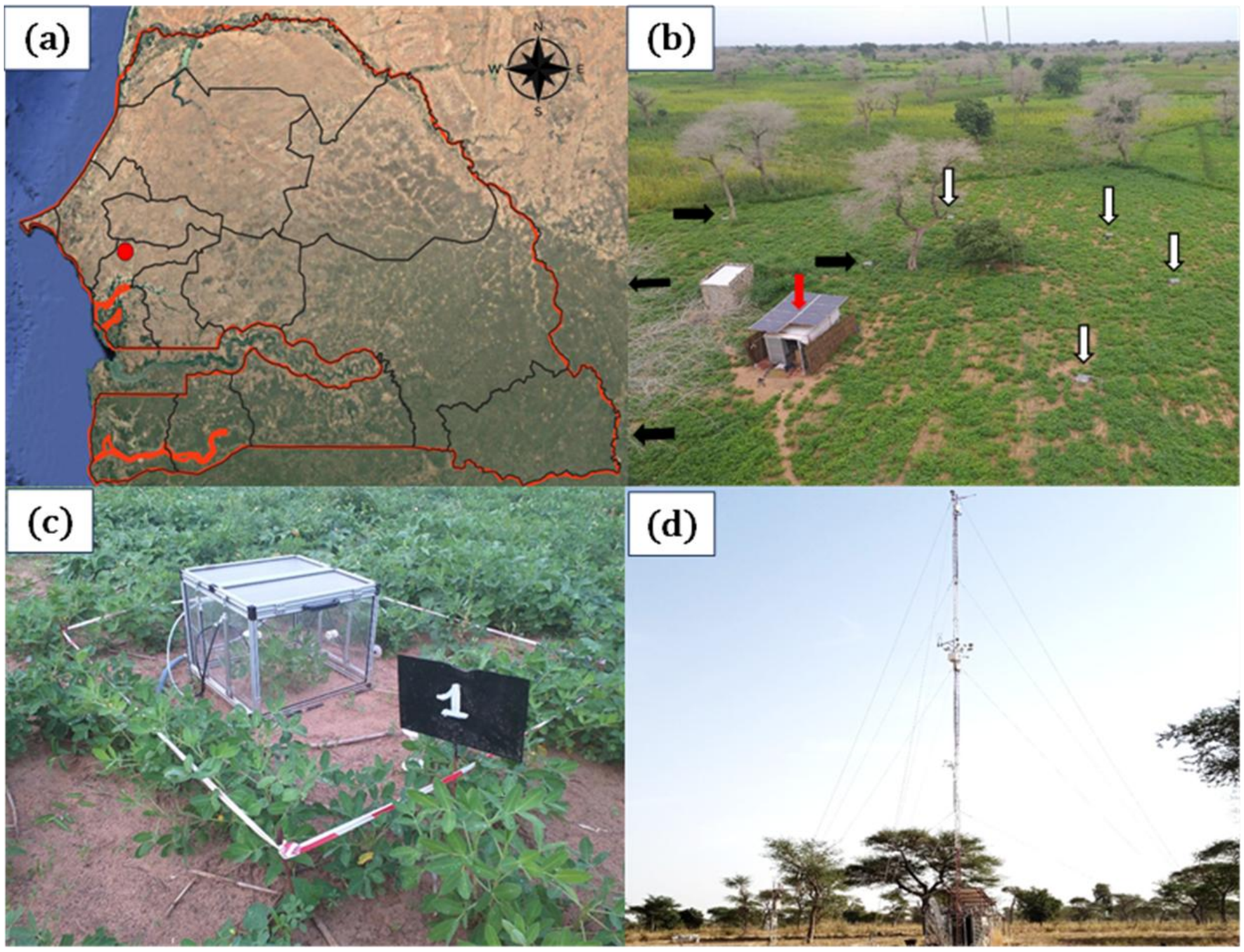

Fig. 1: Study area.

(a) geographical location of Sob, Groundnut basin, Senegal (Map data © Google Earth, 2025), (b) overview (image from the Eddy Covariance tower located in the same bush-field) of the *Faidherbia albida* parkland during the rainy season, depicting groundnut crops with bare soil in the inter-row, *F. albida* trees (defoliated during the rainy season, average height = 13m) and location of the chambers under the Shade of trees (horizontal black arrows; N=4) and in Full sun (vertical white arrows; N=4); The shelter (red arrow) with solar panels is to fit the analyser, automation and batteries (c) automatic chamber enclosing a groundnut plant (during the rainy season) or bare soil (during the dry season), (d) Eddy Covariance (EC) tower (measurement height = 20 m) during the dry season.

### *2.2. Experimental setup*

#### *2.2.1. $CO_2$ flux measurements in automatic chambers*

Continuous net $CO_2$ fluxes ($FCO_2ch$) from soil and groundnut plants were measured over a full phenological year (June 17, 2021 – June 17, 2022) using eight automated static chambers (50×50×50 cm), each enclosing one groundnut plant. Four chambers were installed in full sun (FS), at least 20 m from trees, and four under *F. albida* canopy shade (Sh). The chambers were transparent, custom-built (Duthoit et al., 2020), and installed on metal bases embedded 10 cm into the soil one month prior to measurements.

During the rainy season (June–November), groundnut coexisted briefly with spontaneous weeds until weeding (mid-July), after which chambers contained only groundnut. Post-harvest (early November), chambers remained bare while surrounding plots experienced weed regrowth.

$CO_2$ concentrations were measured at 1 Hz using a Picarro G2508 gas analyser (Picarro Inc., Santa Clara, CA, USA) (Fleck et al., 2013; Reum et al., 2019; Valujeva et al., 2022). A fully automated system was built for sequential half-hour flux measurements (alternating FS and Sh). Measurement duration was 15 min per chamber in the dry season, reduced to 5 min during the rainy season to limit condensation effects.

#### *2.2.2. $CO_2$ flux measurements by Eddy Covariance*

The EC system (Li-COR SMARTFLUX®, including a Gill MasterPro 3D sonic anemometer and a LI-7500 RS open path $CO_2$ and $H_2O$ gas analyser) was mounted at a height of 20 m on a 30m mast, above *F. albida*. It continuously monitored net $CO_2$ exchange from the ecosystem. Raw data were collected at 20 Hz frequency and post-processed from binary files using the advanced mode of the EddyPro® v7.0, with standard corrections and procedures: sonic tilt correction (double rotation), block averaging, covariance maximisation for time lag, and WPL correction (Webb et al., 1980). Quality control followed Foken et al. (2004) and Vickers & Mahrt (1997); random uncertainty was estimated per Finkelstein & Sims (2001). Spectral corrections were applied according to Moncrieff et al. (1997, 2004). Footprints were computed according to Kormann and Meixner (2001), using the FREddyPro R package (Xenakis, 2016), indicated a ~1 ha source area covering the entire field. Gap-filling and flux partitioning were conducted using ReddyProc (Wutzler et al., 2018), applying the daytime partitioning approach of Lasslop et al. (2010).

#### *2.2.3. Ancillary measurements*

Environmental and vegetation variables were monitored continuously throughout the study. Global radiation (Rg) was estimated from photosynthetically active radiation (PAR) using a Skye sensor (averaged over 30-min intervals). NDVI of crops under full sun was recorded semi-hourly

by a calibrated downward-facing sensor installed at 20 m height (Pontailler et al., 2003), processed following Soudani et al. (2012), and used to estimate LAI time series for groundnut, weeds, and cowpea based on end-of-season field LAI measurements in six 15 $m^2$ plots (as in Roupsard et al., 2020).

Rainfall was recorded by an automatic weather station (CR1000 with TE525MM rain gauge, Campbell Scientific), and soil volumetric water content (VWC) and temperature ($T_{soil}$, at 6 cm depth) were monitored using TOMST® TMS-4 sensors, benchmarked prior to field deployment inside and outside the chambers (Wild et al., 2019). Air temperature ($T_{air}$) was recorded inside each chamber at 15 cm above ground, all at 5-min intervals. These measurements contribute to the SoilTemp global database (Lembrechts et al., 2020, 2022).

Groundnut development was tracked weekly by counting leaves in each chamber. Total groundnut LAI (LAIch) was then derived from average single-leaf area and chamber surface.

A detailed description of the data used in this study is provided in Supplement S1 (Table S1.1).

## *2.3. Data processing*

### *2.3.1. Flux calculation*

Net $CO_2$ fluxes ($FCO_2ch$, in µmol $CO_2$ $m^{-2}$ $s^{-1}$) from the chambers were calculated from the linear change in $CO_2$ concentration over time ($\Delta C/\Delta t$) using the Eq.1.

$$\mathbf{FCO_2ch} = \left(\frac{P}{RTk}\right)\left(\frac{V}{A}\right)\left(\frac{\Delta C}{\Delta t}\right) \qquad \textbf{(Eq. 1)}$$

where P is atmospheric pressure (101 325 N $m^{-2}$), R is the ideal gas constant (8.31 N m $mol^{-1}$ $K^{-1}$), $T_k$ is air temperature inside the chamber in Kelvin, V (0.125 $m^3$) is the total system volume (chamber, tubing, analyser cavity, pump, and water trap), and A (0.25 $m^2$) is the chamber footprint. The slope $\Delta C/\Delta t$ was obtained via linear regression (Duthoit et al., 2020).

Mean $FCO_2ch$ values were computed separately for the four replicate chambers in full sun (FS) and under *F. albida* shade (Sh). By convention, negative values indicate net $CO_2$ uptake (photosynthesis), and positive values indicate net $CO_2$ release (respiration).

### *2.3.2. Quality control of chamber-based $CO_2$ flux measurements*

The quality of chamber-based $CO_2$ flux measurements was assessed using the coefficient of determination ($R^2 \geq 0.8$) of the linear increase in $CO_2$ concentration during chamber closure. The minimum detectable flux (MDF) was then calculated following Nickerson (2016) (Eq.2). The MDF defines the flux detection threshold, below which data are considered unreliable due to instrument sensitivity and sampling constraints (Zaman et al., 2021). In this study, the MDF was $\pm$0.0004 µmol $CO_2$ $m^{-2}$ $s^{-1}$.

$$\mathbf{MDF} = \left(\frac{\mathbf{Aa}}{\mathbf{tc}(\sqrt{\mathbf{tc/ps}})}\right)\left(\frac{\mathbf{VP}}{\mathbf{ART}}\right) \quad \textbf{(Eq. 2)}$$

where $A_a$ is the analytical precision of the Picarro analyser (0.6 ppm; Picarro Inc., 2015), tc the closure time (s), $p_s$ the sampling frequency (1 Hz), V the chamber volume, P the atmospheric pressure (101 325 N $m^{-2}$), A the chamber footprint, R the gas constant (8.3 N m $mol^{-1}\cdot K^{-1}$), and T the air temperature in Kelvin.

Following this quality control, fluxes were partitioned (Section 2.3.3) and gap-filled (Section 2.3.4).

### *2.3.3. Partitioning of chamber-based $CO_2$ fluxes*

The net $CO_2$ fluxes ($FCO_2ch$), averaged from four chambers per environment (FS and Sh), were partitioned into two components according to Eq. 3 (Reichstein et al., 2005).

$$\mathbf{FCO_2ch} = \mathbf{Rch} + \mathbf{GPPch} \qquad \textbf{(Eq. 3)}$$

Rch includes heterotrophic respiration (Rh) from soil and other autotrophic respiration (Ra) from groundnut plants and roots of *F. albida* (Ra Groundnut + Ra tree below-ground). Rch is always positive (Rch > 0). GPPch (Gross Primary Productivity) represents the photosynthetic $CO_2$ uptake by the groundnut plants and is negative during the day (GPPch < 0), and zero at night, when $FCO_2ch$ = Rch.

Half-hourly $FCO_2ch$ fluxes were partitioned as follows: (**1**) an Arrhenius-type function (Lloyd & Taylor, 1994) was fitted between nocturnal Rch and $T_{soil}$ during nighttime periods, for each 5-days throughout the time series (Eq. 4). This empirical formulation is based on several key assumptions. First, the relationship between nocturnal respiration and soil temperature is assumed to follow an exponential response, reflecting the temperature sensitivity of respiration processes. Second, the model assumes temporal stability of the respiration–temperature relationship between night and day, allowing diurnal respiration to be extrapolated from fitted parameters in Eq.4 and daytime $T_{soil}$. Third, we assumed that no abrupt changes in substrate availability or soil moisture occur between day and night — conditions that could otherwise disrupt the temperature–respiration relationship. Third, it is assumed that no abrupt changes in substrate availability or soil moisture occur between night and day — conditions that could otherwise decouple respiration rates from temperature. These assumptions are widely applied in $CO_2$ flux partitioning approaches (Reichstein et al., 2005; Lasslop et al., 2010). (**2**) Diurnal Rch was estimated by applying the Lloyd & Taylor function, previously calibrated on nocturnal data, to the corresponding daytime $T_{soil}$ measurements for each 5-day interval. (**3**) GPPch was subsequently derived as the residual component of the net $CO_2$ flux during the day, according to:

$$\textbf{nocturnal Rch} = \mathbf{R_{ref}} \cdot \mathbf{exp}\left[\mathbf{E_0}\left(\frac{1}{\mathbf{T_{ref}-T_0}} - \frac{1}{\mathbf{T_{soil}-T_0}}\right)\right] \quad \textbf{(Eq. 4)}$$

where $R_{ref}$ (µmol $CO_2$ $m^{-2}$ $s^{-1}$) is a fitted parameter representing the base respiration at the reference temperature [$T_{ref}$ (K), (set at 288.15 K)]. $E_0$ (K) is the temperature sensitivity (set at 250 K), $T_{soil}$ (K) the soil temperature (K), and $T_0$ (K) is kept constant at 231.13 K, according to Lloyd & Taylor (1994).

$$\textbf{GPPch} = \textbf{diurnal FCO}_2\textbf{ch} - \textbf{diurnal Rch} \quad \textbf{(Eq. 5)}$$

where diurnal $FCO_2ch$ and diurnal Rch represent the daytime net $CO_2$ fluxes and respiration in µmol $CO_2$ $m^{-2}$ $s^{-1}$, respectively.

### *2.3.4. Gap-filling procedure*

Missing Rch data were gap-filled using the model derived from Eq. 4 (Lloyd & Taylor, 1994). Prior to gap-filling GPPch, raw data were standardised by LAI to reduce variability between chambers due to differences in leaf surface area (Eq. 6). A light-response model was then fitted to the standardised GPPch data, every 5-day period, to gap-fill missing values. The model is based on a rectangular hyperbolic function that describes the relationship between photosynthetic $CO_2$ uptake and incoming global radiation (Rg) (Eq. 7). It corresponds to a Michaelis–Menten-type light-response curve, commonly used in ecosystem carbon exchange studies (Falge et al., 2001; Lasslop et al., 2010).

$$\textbf{GPPch.stand} = \frac{\textbf{GPPch}}{\textbf{LAIch}} * \textbf{LAI.field} \quad \textbf{(Eq. 6)}$$

where GPPch.stand (µmol $CO_2$ $m^{-2}$ $s^{-1}$) is the standardised GPPch. LAIch and LAI.field ($m^2$ leaves $m^{-2}$ soil) represent the groundnut LAI inside the chambers and the groundnut + weeds +cowpea LAI for the whole field, respectively.

$$\textbf{GPP} = \frac{\boldsymbol{\alpha\beta}\textbf{Rg}}{\boldsymbol{\alpha}\textbf{Rg} + \boldsymbol{\beta}} \quad \textbf{(Eq. 7)}$$

where α (µmol $CO_2$ $J^{-1}$) represents the light use efficiency of the groundnut plants inside the chambers, and refers to the initial slope of the light-response curve, β (µmol $CO_2$ $m^{-2}$ $s^{-1}$) is the maximum $CO_2$ uptake rate by the groundnut plants at light saturation, and Rg the global radiation (W $m^{-2}$).

### *2.3.5. Comparing chamber-based (Ch) and Eddy Covariance (EC) methods*

Chamber measurements were upscaled to field-level $CO_2$ fluxes and compared with EC-derived fluxes. Before comparison, a correction was applied (Eq. 6) to account for differences in LAI between chambers (LAIch) and the field (LAI.field), due to the presence of cowpea and weeds in the field but not in the weeded chambers.

Upscaling considered tree cover, with FS and Sh chamber fluxes weighted at 90% and 10%, respectively. Rch.stand and GPPch.stand, representing chamber-based respiration and

photosynthesis at field scale. These fluxes were compared, on a half-hourly basis, to EC-derived Reco.EC and GPP.EC (S3, Table S3.1). The November–December transition period was excluded due to weed-driven uncertainties after groundnut harvest.

During the rainy season (F. albida leafless), GPP.EC represented ground vegetation (groundnut, cowpea, weeds), while Reco.EC included autotrophic respiration from all vegetation (including trees), and heterotrophic respiration (Reco.EC = Ra tree below-ground + Ra tree above-ground + Ra groundnut + Ra cowpea + Ra weeds + Rh). Rch.stand could not be fully upscaled to the field due to uncertainty in its partitioning between Ra and Rh. Rch.stand accounted only for Ra tree below-ground, Ra groundnut, and Rh.

In the dry season (leafy trees, bare soil), GPP.EC reflected tree photosynthesis only (GPP tree), while GPPch.stand was nil. Reco.EC included Ra tree (above- and below-ground) and Rh. Rch.stand, measured on bare soil represented only Ra tree below-ground + Rh.

#### *2.3.6. Contribution of trees to full ecosystem respiration and photosynthesis*

During the dry season, when the trees (*F. albida*) maintained their foliage, a comparison between chamber and EC measurements allowed for the estimation of the contribution of the above-ground tree compartments to total ecosystem respiration (S3, Table S3.1). Based on this estimate, total tree respiration (Ra tree) was then calculated under the assumption that the tree root systems (Ra tree below-ground) represent ⅓ of the above-ground biomass (Jackson et al. 1996). Given the GPP measured during the dry season was equivalent to GPP of trees (GPP trees) from EC measurements, the carbon use efficiency of the trees (CUE tree) was then calculated (S3, Table S3.1). The resulting CUE value was assessed to determine whether it approximated the typical value of 0.5, which is often used as a default in ecosystem models (Zhou et al., 2019; 2020).

#### *2.3.7. Net annual C budget at the ASPS scale*

The annual C budget of $CO_2$ fluxes was estimated for chambers and EC measurements in Mg C-$CO_2$ $ha^{-1}$. The chambers $CO_2$ fluxes budgets were obtained by calculating the annual sum of the net $CO_2$ flux measurements and then weighting with the tree cover rate (10% for the Sh, 90% for the FS). These annual budgets for the field are considered apparent, as they do not account for the biomass exported from the field after the harvest, the decomposition of which therefore escaped both the chambers and the EC. Additionally, the inputs and the outputs of fecal matter resulting from livestock wandering during the dry season were not quantified and are therefore neglected. The objective here is to compare two approaches at different scales using apparent net C budgets, rather than to provide an absolute C budget.

### 2.4. *Statistical analyses*

Statistical analyses were performed using the R software (R. Core Team, 2023). To compare the mean values of climatic parameters between the FS and Sh situations, a non-parametric Mann-Whitney test was used when both the normality (shapiro.test) and the homogeneity of the variance (Levene Test, R package 'Car'; Fox et al., 2023) were not confirmed. This approach was similarly applied to compare the seasonal dynamics of $CO_2$ fluxes between FS and Sh, as well as between the chamber-based and Eddy Covariance (EC) methods. Means and standard deviations were computed using the 'skim' function from the R package 'skimr' (Waring et al., 2022).

Respiration (Rch) (Eq. 4) and GPP (GPPch) models (Eq. 7) were fitted using non-linear least squares regression, implemented in the library in R 'nls.multstart' (Padfield et al., 2025). For the GPPch model, parameters α and β with non-significant p-values were removed, and then the remaining values were interpolated and smoothed using a 'spline' function from the 'zoo' library in R (Zeileis et al., 2024). Ordinary least-square linear regressions were fitted between the measured and the modeled values derived from. Model performance of Eq. 4 and Eq. 7 was evaluated by fitting ordinary least-square linear regressions between the measured and the modeled values using $R^2$, root mean square error (RMSE), and the bias metrics. Given that the primary objective of these equations was to accurately reproduce the seasonal dynamics of the $CO_2$ fluxes to fill gaps in data, particular emphasis was placed on $R^2$, with a higher value reflecting a better fit of the model to the measurements.

Correlation analysis was conducted between chamber $CO_2$ fluxes ($FCO_2ch$, Rch, GPPch) and soil temperature ($T_{soil}$, °C), air temperature ($T_{air}$, °C), VWC, the leaf area index of groundnut plants in the chambers (LAIch), and the fitted parameters for respiration — $R_{ref}$ — and photosynthesis — α and β. This analysis was performed using the 'cor.test' function from the 'stats' package in R (Lüdecke et al., 2021), applying the Spearman method.

The threshold of the daily mean soil temperature ($T_{soil}$, °C) at which the cumulative daily respiration (Rch, g C-$CO_2$ $m^{-2}$ $d^{-1}$) began to decline was determined using segmented regression from the R package 'segmented' (Muggeo, 2003). The associated uncertainty (standard error) of this estimate was evaluated through a bootstrap procedure.

# 3. Results

## *3.1. Microclimatic conditions*

During the experiment, the cumulative rainfall was 550 mm, which was representative of the interannual average. Precipitations were lowest in July and highest between August and September, a period that typically corresponds to the peak of the rainy season (Fig. 2a). Global radiation ranged between 5.8 and 32.4 MJ $m^{-2}$ $d^{-1}$ (data not shown). The daily mean VWC in the chambers showed significant variation, ranging from 1% at the end of the dry season to a maximum of 30% during the rainy season (Fig. 2a). While VWC was similar during the rainy season, it remained consistently higher in FS than in Sh throughout the dry season ($p < 0.05$), which was unexpected. However, it should be noted that the last rain of October 2021 recharged the FS chambers more effectively, likely due to foliage rainfall interception by *F. albida* which had just put on leaves at that time, potentially explaining this discrepancy in VWC.

Within the chamber, the daily mean $T_{soil}$ ranged from 26°C in April to 37.5°C at the end of the dry season (Fig. 2b), while $T_{air}$ varied between 23.7°C and 35.5°C (Fig. 2c). However, during instantaneous daily peaks, $T_{soil}$ could exceed 45°C in May (data not shown). As expected, both daily mean $T_{soil}$ and $T_{air}$ were significantly higher in FS compared to Sh situations ($p < 0.05$), with $T_{soil}$ and $T_{air}$ averaging respectively 1°C and 0.5°C lower under the tree canopy.

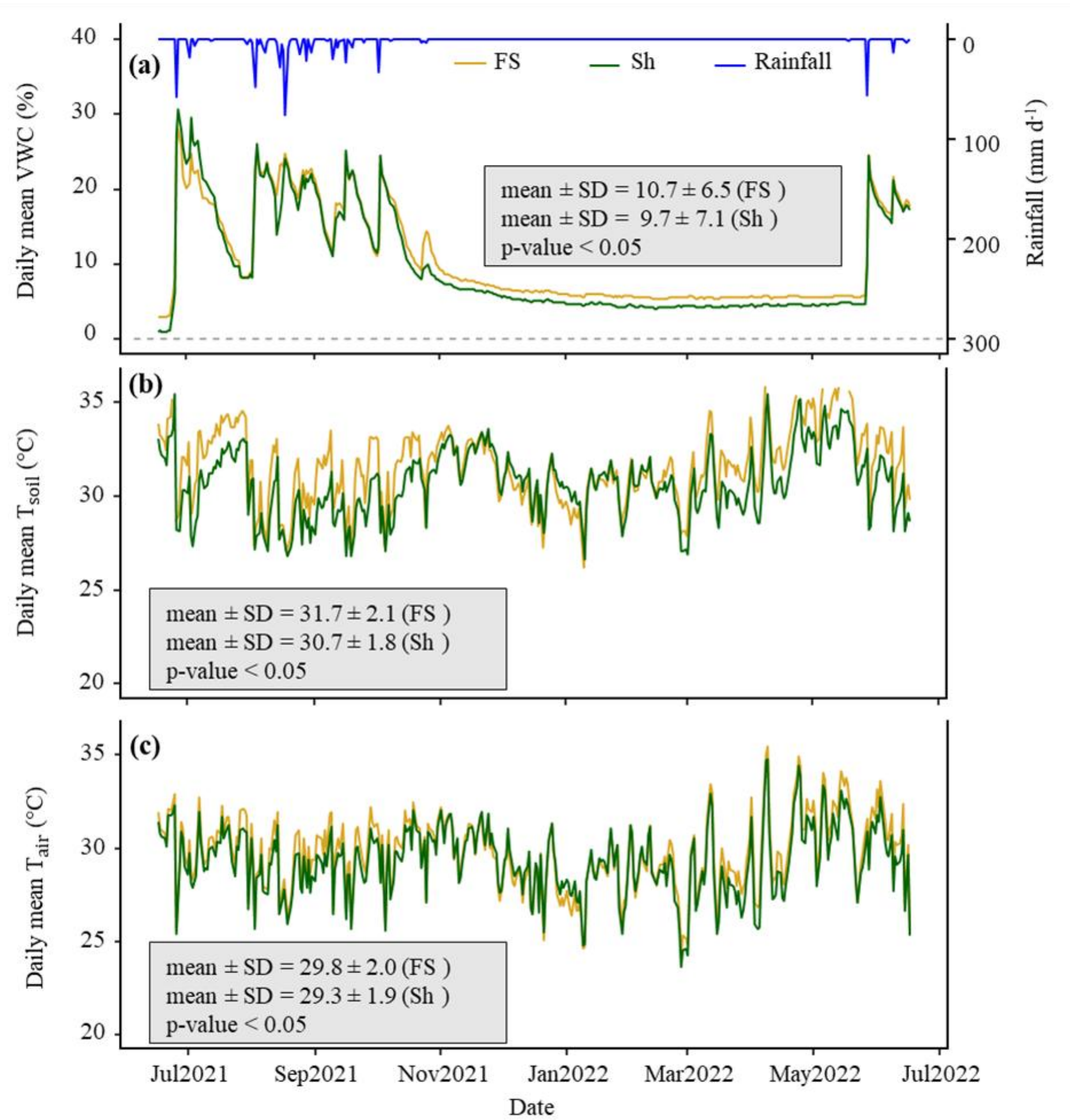

Fig. 2: One-year time series of daily average microclimatic parameters measured inside chambers.

(a) volumetric soil water content (VWC) at a depth of 6 cm (%). (b) soil temperature ($T_{soil}$) at a depth of 6 cm (°C), (C) air temperature ($T_{air}$) at a height of 15 cm (°C). The blue line depicts the daily rainfall (mm $d^{-1}$) throughout the year. FS: Full sun chambers; Sh: Shaded chambers. Mean and SD represent respectively the mean value and the standard deviation. The p-value indicates the probability associated with the statistical test, assessing the differences in means between FS and Sh with the significance level α set to 0.05.

### *3.2. Modeling the chamber-based total respiration (Rch) and photosynthesis (GPPch)*

#### *3.2.1. Dynamics of references respiration, light use efficiency, and maximum $CO_2$ uptake rate at light saturation ($R_{ref}$, α., and β)*

The reference respiration ($R_{ref}$) showed comparable seasonal dynamics both at a distance from the trees (FS) and under the tree canopies (Sh) (S2, Fig. S2.2). In both situations, $R_{ref}$ showed strong variability during the rainy season, peaking in September 2021 at 2.4 μmol $CO_2$ $m^{-2}$ $s^{-1}$ for FS and 2.9 μmol $CO_2$ $m^{-2}$ $s^{-1}$ for Sh (S2, Table S2.1). In contrast, during the dry season — from November 3, 2021 (after harvest) until the onset of the following rainy season (June 2022) — $R_{ref}$ values dropped both for FS and Sh, averaging 0.3 ± 0.5 μmol $CO_2$ $m^{-2}$ $s^{-1}$ for FS and 0.5 ± 0.6 μmol $CO_2$ $m^{-2}$ $s^{-1}$ for Sh. This represents a reduction by a factor of 8 for FS and 6 for Sh compared to the rainy season. The mean annual $R_{ref}$ values were significantly higher under Sh than in FS, with value approximately 1.5 times greater (S2, Table S2.1).
Regarding GPP in chambers, the light use efficiency (α) and the maximum $CO_2$ uptake by groundnut plants in the chambers (β), also reached their maximum during the peak of the rainy season (S2, Fig. S2.3, a and b). The maximum value of α reached 0.2 μmol $CO_2$ $J^{-1}$ in FS and 0.3 μmol $CO_2$ $J^{-1}$ in Sh (S2, Table S2.1). Similarly, the maximum values of optimum $CO_2$ uptake rate at light saturation (β) were 40.2 μmol $CO_2$ $m^{-2}$ $s^{-1}$ for FS and 42.8 μmol $CO_2$ $m^{-2}$ $s^{-1}$ for Sh (S2, Table S2.1). In the dry season, when photosynthetic activity ceased in the chambers, both α and β were assumed to be nil (S2, Fig. S2.3, a and b). On average, α and β were significantly higher in Sh than in FS, by a factor of 1.7 and 1.2, respectively (S2, Table S2.1). We noted that the decline in photosynthetic activity of the groundnut crop occurred earlier and rapidly at a distance from the trees (FS), as reflected by the sharply observed recession of α and β in FS.

#### *3.2.2. Dynamics of nocturnal respiration in chambers*

The averaged nocturnal respiration (nocturnal Rch) calculated from the measurements across each treatment (FS and Sh), showed similar seasonal patterns (Fig. 3, a and c). Following the first rains, Rch values increased dramatically, with a nocturnal 'Birch effect' — a sudden pulse of $CO_2$ release following soil rewetting — observed to be more pronounced under Sh compared to FS, approximately by a factor of 2. At the peak of the rainy season (September), the maximum nocturnal Rch values reached approximately 6.0 μmol $CO_2$ $m^{-2}$ $s^{-1}$ in FS and 9.0 μmol $CO_2$ $m^{-2}$ $s^{-1}$ in Sh (Fig. 3, a and c). Thereafter, nocturnal Rch declined well before the groundnut harvest along with the rainfall spacing and the groundnut crop senescence (data not shown). During the dry season nocturnal Rch continued to decrease, with maximum values around 1.0 μmol $CO_2$ $m^{-2}$ $s^{-1}$ in FS and 2.0 μmol $CO_2$ $m^{-2}$ $s^{-1}$ in Sh (Fig. 3, a and c).

The modeled nocturnal Rch values closely matched the measured nocturnal Rch values (mean across four chambers per treatment), as indicated by the model performance metrics ($R^2 = 0.9$, with bias and RMSE values of 0.3 and 0.5 µmol $CO_2$ $m^{-2}$ $s^{-1}$, respectively, for FS; $R^2 = 0.7$, with bias and RMSE values of 0.4 and 0.6 µmol $CO_2$ $m^{-2}$ $s^{-1}$, respectively, for Sh) (Fig. 3, b and d). Similarly, the daily mean modeled values also fitted well with the measured values, with FS showing 0.9 ± 0.9 µmol $CO_2$ $m^{-2}$ $s^{-1}$ (modeled) and 1.2 ± 1.2 µmol $CO_2$ $m^{-2}$ $s^{-1}$ (measured), while Sh recorded 1.4 ± 0.9 µmol $CO_2$ $m^{-2}$ $s^{-1}$ (modeled) and 1.5 ± 1.2 µmol $CO_2$ $m^{-2}$ $s^{-1}$ (measured). Given the close match between the measured and modeled values, the fitted model parameters were used subsequently to fill data gaps and estimate diurnal Rch values, as presented in Fig. 4, a and c.

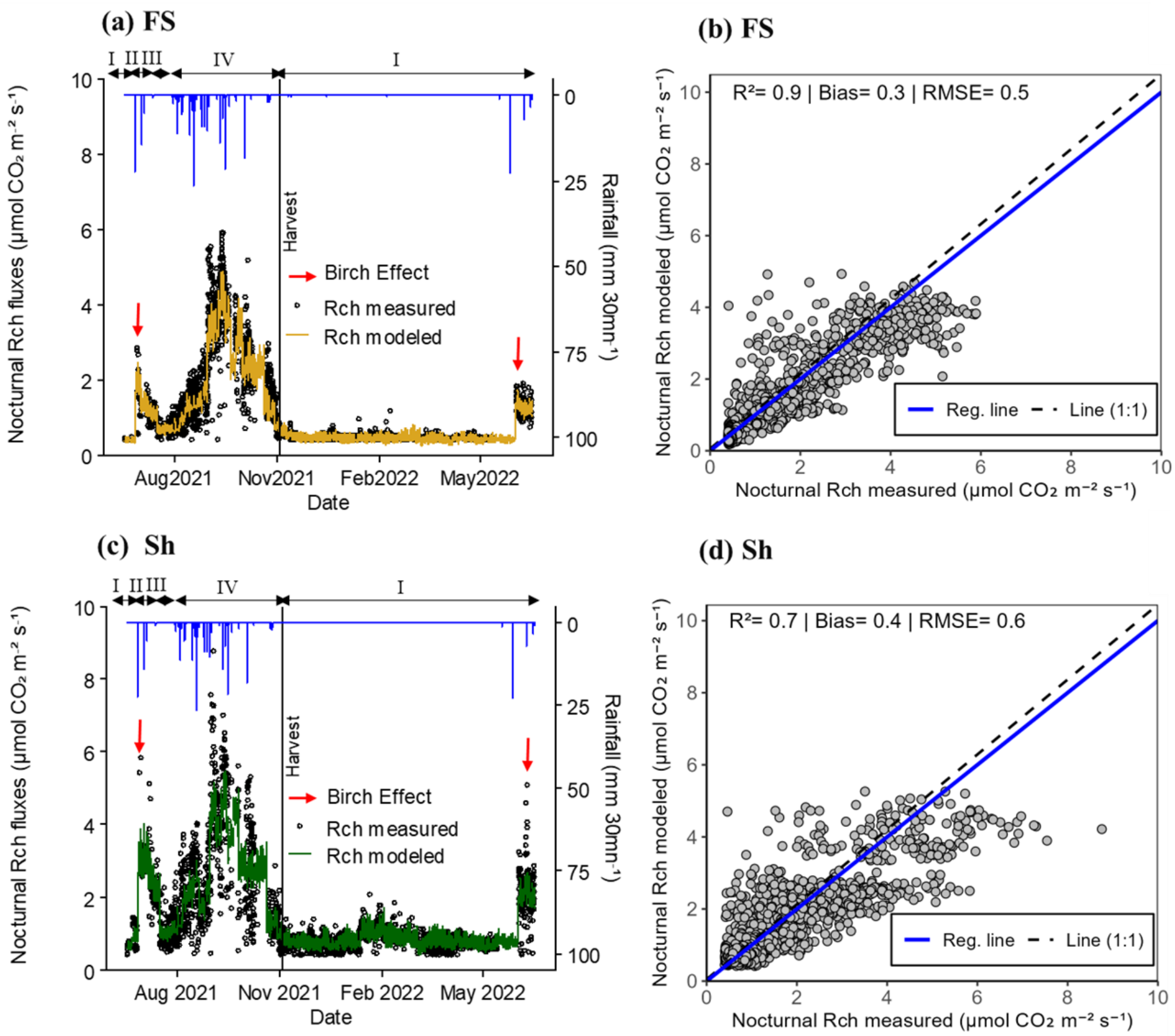

Fig. 3: Dynamics of instantaneous nocturnal $CO_2$ fluxes in chambers in Full sun (FS; a and b) and Shaded (Sh; c and d) environments (data filtered based on $R^2$ of the $CO_2$ variation over the time of chamber closure and Minimum Detectable Flux, Eq.2).

(a) and (c): measured nocturnal respiration in chambers (Rch: black dots; average of measurements in 4 chambers per location) vs. modeled (coloured line). The vertical black line indicates the harvest date of groundnuts inside the chambers. The red arrows indicate the 'Birch' effect and the blue line represents the rainfall (mm $30mn^{-1}$). Roman numerals (above the black arrows) refer to vegetation conditions prevailing inside the chambers, i.e. (I) bare soil, (II) weeds, (III) weeds + groundnuts, and (IV) groundnuts only.
(b) and (d): scatter plot between measured and modeled nocturnal Rch. The solid blue line indicates the regression line and the dashed black one the (1:1) line RMSE and bias are expressed as fluxes (in µmol $CO_2$ $m^{-2}$ $s^{-1}$). Each point represents the mean value from 4 chambers within the FS or Sh environments.

*3.2.3. Dynamics of daytime fluxes in chambers*

The measured GPPch.stand, as well as GPP modeled with Eq. 6, showed similar seasonal dynamics inFS and Sh (Fig. 4, a and c). The fluxes peaked during the rainy season (Fig. 4a and c), coinciding with periods of vigorous vegetative growth characterised by a high leaf area index (LAIch) of groundnut plants within the chambers (S2, Fig. S2.1). The maximum calculated and standardised GPPch values reached -50 µmol $CO_2$ $m^{-2}$ $s^{-1}$ for FS and -37 µmol $CO_2$ $m^{-2}$ $s^{-1}$ for Sh. As expected, these fluxes were nil during the dry season when the soil was bare (Fig. 4, a and c).
The modeled GPPch values closely followed the same trends as the calculated values, although model performance was slightly better for FS ($R^2 = 0.7$ with bias and RMSE values of 4.2 and 6.1 µmol $CO_2$ $m^{-2}$ $s^{-1}$, respectively) compared to Sh ($R^2 = 0.6$ with bias and RMSE values of 6.1 and 5.6 µmol $CO_2$ $m^{-2}$ $s^{-1}$, respectively) (Fig. 4, b and d).
The calculated diurnal respiration values (diurnal Rch calculated) for FS and Sh revealed a 'Birch effect' similar to that observed during the night, though slightly more pronounced under Sh by a factor of 1.2. Diurnal Rch values increased significantly during the rainy season, reaching a maximum of 6.0 µmol $CO_2$ $m^{-2}$ $s^{-1}$ for both FS and Sh (Fig. 4, a and c). In the dry season, on bare soil, these values declined, with maximum respiration reaching only 0.5 µmol $CO_2$ $m^{-2}$ $s^{-1}$ for both situations (FS and Sh) (Fig. 4, a and c).

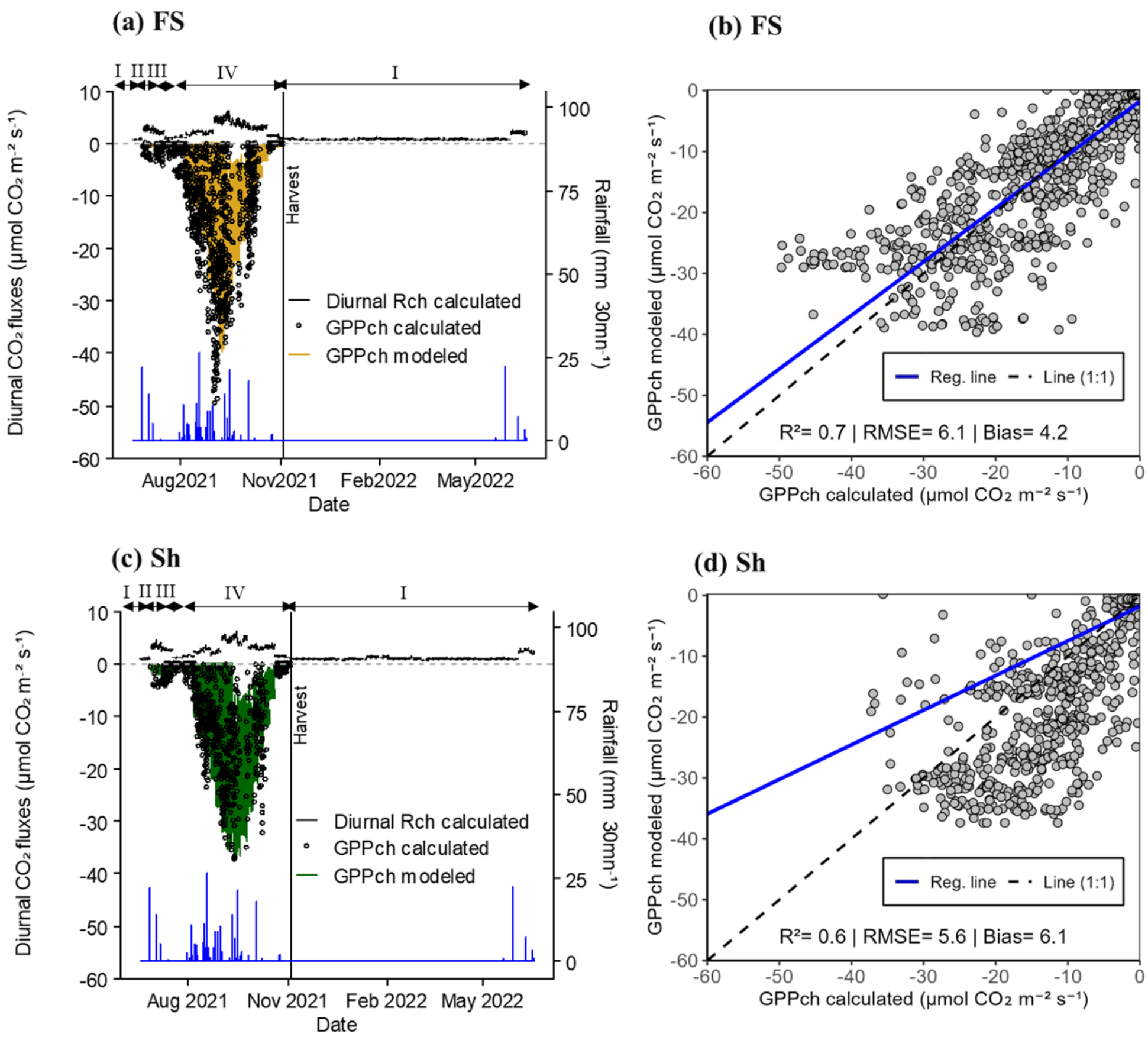

Fig. 4: Dynamics of instantaneous diurnal $CO_2$ fluxes in chambers in Full sun (FS; a and b) and Shaded (Sh; b and d) environments (filtered based on $R^2$ of the $CO_2$ variation over the time closure in FS and Sh and Minimum Detectable Flux, Eq.2).

(a) and (c): non-gap-filled diurnal Rch calculated (black line, positive values; average of measurements in 4 chambers per location) and GPPch calculated from Eq.5 then standardised for LAI (black dots, negative values) and modeled (coloured line, negative values). The vertical black line indicates the harvest date of groundnuts inside the chambers and the blue line represents the rainfall (mm $30mn^{-1}$). Roman numerals (above the black arrows) refer to conditions prevailing inside the chambers, i.e., (I) bare soil, (II) weeds, (III) weeds + groundnuts, and (IV) groundnuts.
(b) and (d): scatter plot between calculated and modeled GPPch. The solid blue line indicates the regression line and the dashed black one the (1:1) line. RMSE and bias are expressed as fluxes (in µmol $CO_2$ $m^{-2}$ $s^{-1}$). Each point represents the mean value from 4 chambers within the FS or Sh environments.

### 3.3. *Dynamics of daily cumulative $CO_2$ fluxes in chambers*

The seasonality of daily cumulative of GPPch.stand showed similar dynamics between FS and Sh, with higher variability during the rainy season than during the dry season (Fig. 5). Daily total Rch peaked during the rainy season at 5.1 g C-$CO_2$ $m^{-2}$ $d^{-1}$ for FS and 5.4 g C-$CO_2$ $m^{-2}$ $d^{-1}$ for Sh, while the maximum GPPch.stand values were comparable at around -15.0 g C-$CO_2$ $m^{-2}$ $d^{-1}$ for both FS and Sh (Table 1; S2, Fig. S2.4, a, b, c, and d). In the dry season, Rch decreased (Fig. 5), averaging 0.5 g C-$CO_2$ $m^{-2}$ $d^{-1}$ for FS and 1.0 g C-$CO_2$ $m^{-2}$ $d^{-1}$ for Sh. GPPch declined well before harvest (senescence) and remained nil during the dry season (Fig. 5). During the rainy season $FCO_2$ch peaked at around 11.0 g C-$CO_2$ $m^{-2}$ $d^{-1}$ for FS and Sh (Fig. 5) (Table 1; S2, Fig. S2.4, e and f), while $FCO_2$ch values were the same as Rch during the dry season. In absolute terms, the mean Rch and GPPch were significantly higher under Sh as compared to FS, by factors of 1.3 and 1.2, respectively. Conversely, the mean $FCO_2$ch was significantly higher under FS (0.4 g C-$CO_2$ $m^{-2}$ $d^{-1}$) than under Sh (0.2 g C-$CO_2$ $m^{-2}$ $d^{-1}$) (Table 1).

The annual cumulative Rch values were 392.8 g C-$CO_2$ $m^{-2}$ for FS and 574.5 g C-$CO_2$ $m^{-2}$ for Sh. The GPPch fluxes reached -539.5 g C-$CO_2$ $m^{-2}$ for FS and -632.6 g C-$CO_2$ $m^{-2}$ for Sh. Annual net cumulative C exchange ($FCO_2$ch) were -146.7 g C-$CO_2$ $m^{-2}$ in FS and -58.1 g C-$CO_2$ $m^{-2}$ in Sh.

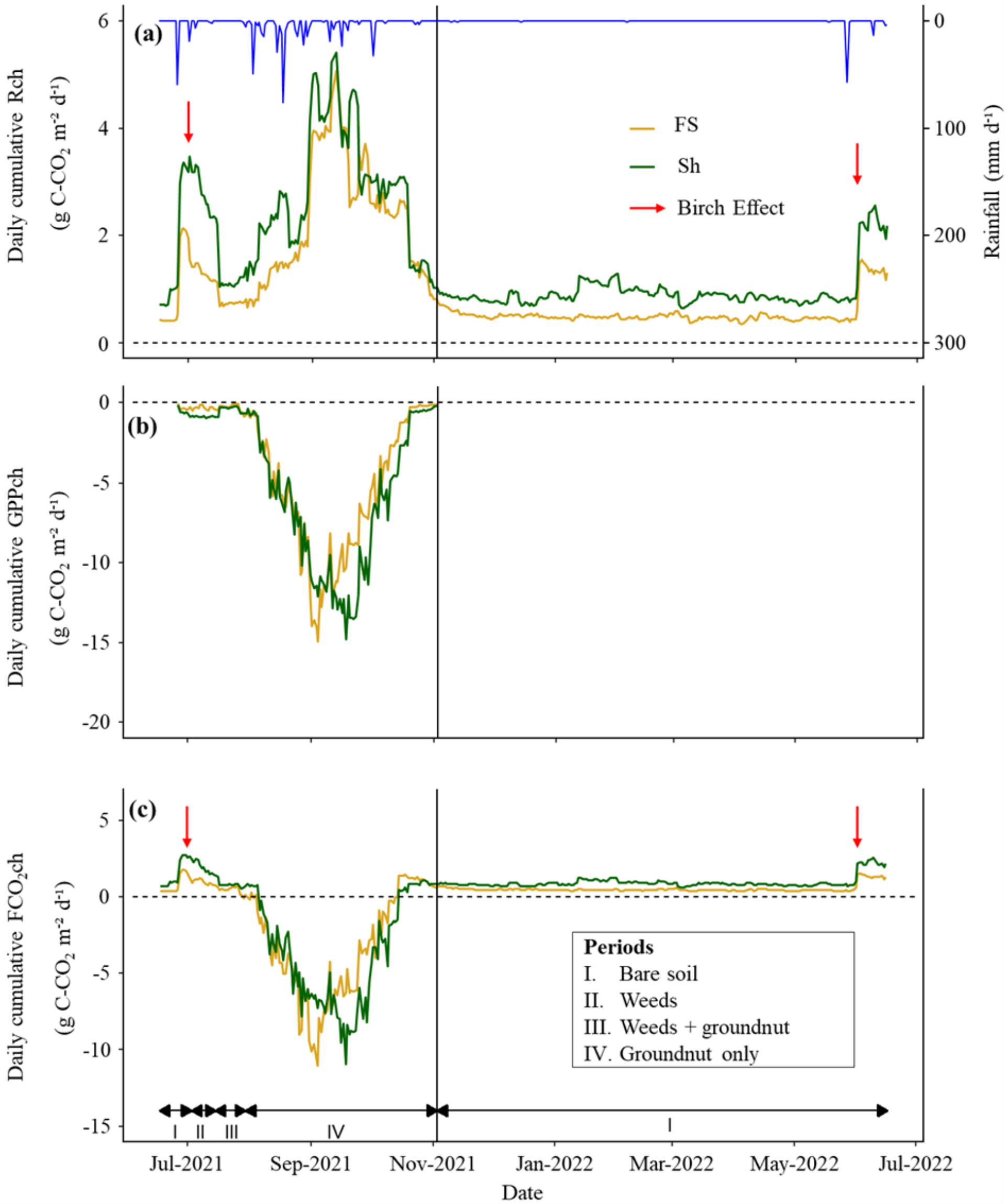

Fig. 5: Seasonal dynamics of daily gap-filled cumulative fluxes (in gC-$CO_2$ m$^{-2}$ d$^{-1}$) in chambers.

(a) soil+crop respiration (Rch), (b) photosynthesis (GPPch, standardised for LAI) and (c) net $CO_2$ exchange ($FCO_2$ch). The yellow and green solid lines compare the FS and Sh environments, respectively. The vertical black line indicates the harvest date of groundnuts inside the chambers. The blue line depicts the daily cumulative rainfall (mm d$^{-1}$) throughout the rainy season, and the red arrow indicates the 'Birch' effect. Roman numerals (above the black arrows) in (a) and (c) refer to the prevailing conditions inside the chambers: (I) bare soil, (II) weeds, (III) weeds + groundnuts, (IV) groundnuts.

Table 1: Comparison of daily cumulative and gap-filled chamber $CO_2$ fluxes (Rch, GPPch standardised for LAI, and $FCO_2$ch in g C-$CO_2$ $m^{-2}$) in the FS and Sh condition.

| | Annual sum | Daily Mean ±SD | Min | Max | Mann-Whitney test |
|---|---|---|---|---|---|
| (g C-$CO_2$ $m^{-2}$) | .$yr^{-1}$ | .$d^{-1}$ | .$d^{-1}$ | .$d^{-1}$ | |
| **Rch** | | | | | |
| FS | 392.8 | 1.1 ± 0.9 | 0.4 | 5.1 | * |
| Sh | 574.5 | 1.6 ± 1.1 | 0.6 | 5.4 | |
| **GPPch** | | | | | |
| FS | -539.5 | -4.1 ± 4.3 | < -0.1 | -14.9 | * |
| Sh | -632.6 | -4.8 ± 4.6 | < -0.1 | -14.8 | |
| **$FCO_2$ch** | | | | | |
| FS | -146.7 | -0.4 ± 2.4 | -11.0 | 1.8 | * |
| Sh | -58.1 | -0.2 ± 2.7 | -10.9 | 2.8 | |

Annual sum corresponds to the annual cumulative fluxes (g C-$CO_2$ $m^{-2}$ $yr^{-1}$). Mean, SD, Min, and Max represent respectively the mean, standard deviation, minimum, and maximum values at the daily scale (g C-$CO_2$ $m^{-2}$ $d^{-1}$). Asterisks (*) indicate the p-values from the Mann-Whitney test, used to assess differences in mean between FS and Sh ($p < 0.05$). Positive values indicate $CO_2$ emissions, while negative values represent $CO_2$ uptake.

### *3.4. Drivers of daily respiration and photosynthesis in chambers*

The chamber-based daily cumulative respiration (Rch) and GPPch showed significant and positive correlations with the leaf area index (LAIch), both at a distance from the trees (FS) and under the trees (Sh) (Table 2). The influence of LAIch on GPPch was stronger (r = 0.86 for FS and Sh) than its influence on Rch (r = 0.60 for FS; r = 0.69 for Sh). Soil VWC was also positively correlated with Rch and GPPch, both in FS and Sh. However, the influence of soil VWC on Rch was stronger under Sh compared to FS, while its influence on GPPch was similar in both situations (FS and Sh). Soil temperature showed weak negative correlations with Rch (in FS and Sh) and with GPPch (only in Sh). Finally, no significant correlations were found between $T_{air}$, and any of the $CO_2$ fluxes (Table 2).

Table 2: Spearman correlation matrix based on daily cumulative and gap-filled $CO_2$ fluxes from full year chamber measurements (g C-$CO_2$ $m^{-2}$ $d^{-1}$) with microclimatic parameters.

| Parameters | Condition | Rch | GPPch |
|---|---|---|---|
| $T_{soil}$ | FS | -0.25 *** | ns |
| | Sh | -0.28 *** | -0.38 *** |
| $T_{air}$ | FS | ns | ns |
| | Sh | ns | ns |
| VWC | FS | 0.51 *** | 0.75 *** |
| | Sh | 0.78 *** | 0.75*** |
| LAIch | FS | 0.60 *** | 0.86 *** |
| | Sh | 0.69 *** | 0.86 *** |

Spearman correlation coefficients between daily cumulative and gap-filled $CO_2$ flux components (Rch and GPPch, with GPPch in absolute terms) and daily mean microclimatic parameters in full sun (FS) and shaded chambers (Sh). $T_{soil}$ (°C) is the daily mean soil temperature at 6 cm depth, $T_{air}$ (°C) the daily mean air temperature at 15 cm height, VWC (%) the daily mean volumetric water content (VWC, %), and LAIch ($m^{-2}$ leaf $m^{-2}$ soil) the chamber leaf area index value for a given day. Significance levels are indicated by (***) for $p < 0.001$; ns denotes a non-significant correlation ($p > 0.05$)

### *3.5. Comparison of respiration and GPP measurements between chambers (Ch) and Eddy Covariance (EC) methods*

The chamber-based daily total $CO_2$ fluxes, gap-filled and weighted according tree cover were compared with the fluxes obtained using the EC method (Fig. 6).

During the rainy season, both total respiration and GPPshowed comparable dynamics between the two methods, with synchronised peaks and higher variability compared to the dry season (Fig. 6). The maximum value of Reco.EC, peaked at 13.5 g C-$CO_2$ $m^{-2}$ $d^{-1}$ (Table 3). The initial value of Rch.stand was comparable to Reco.EC but peaked only at 5.1 g C-$CO_2$ $m^{-2}$ $d^{-1}$ (Table 3), meaning a third of the peak of Reco.EC. The maximum GPP, was -14.3 g C-$CO_2$ $m^{-2}$ $d^{-1}$ and -14.6 g C-$CO_2$ $m^{-2}$ $d^{-1}$ for GPP.EC and GPPch.stand, respectively (Table 3). This indicates that the LAI-based standardisation and upscaling approach were realistic, at least up to the peak of groundnut growth.

On average, Reco.EC was significantly higher than Rch.stand, by a factor of 2.3. GPP.EC was also significantly higher than GPPch.stand, but only by a factor of 1.2 (Table 3).

During the dry season, Reco.EC and Rch.stand gradually decreased. The values for Reco.EC remained higher than for Rch.stand, which was fairly consistent with the contribution of the Ra tree above-ground compartment, even if this difference seemed to disappear at the end of the dry season (Fig. 6). The measured 'Birch effect' was highest for Rch.stand in 2021, but was the opposite in 2022 due to a system failure at the beginning of the rainy season. The maximum value of GPP.EC reached -2.4 g C-$CO_2$ $m^{-2}$ $d^{-1}$ when the trees were at their maximum of foliage, after harvest and while weeds were still present in the field. However, after the harvest, chamber photosynthesis (GPPch.stand) was nil (Table 3).

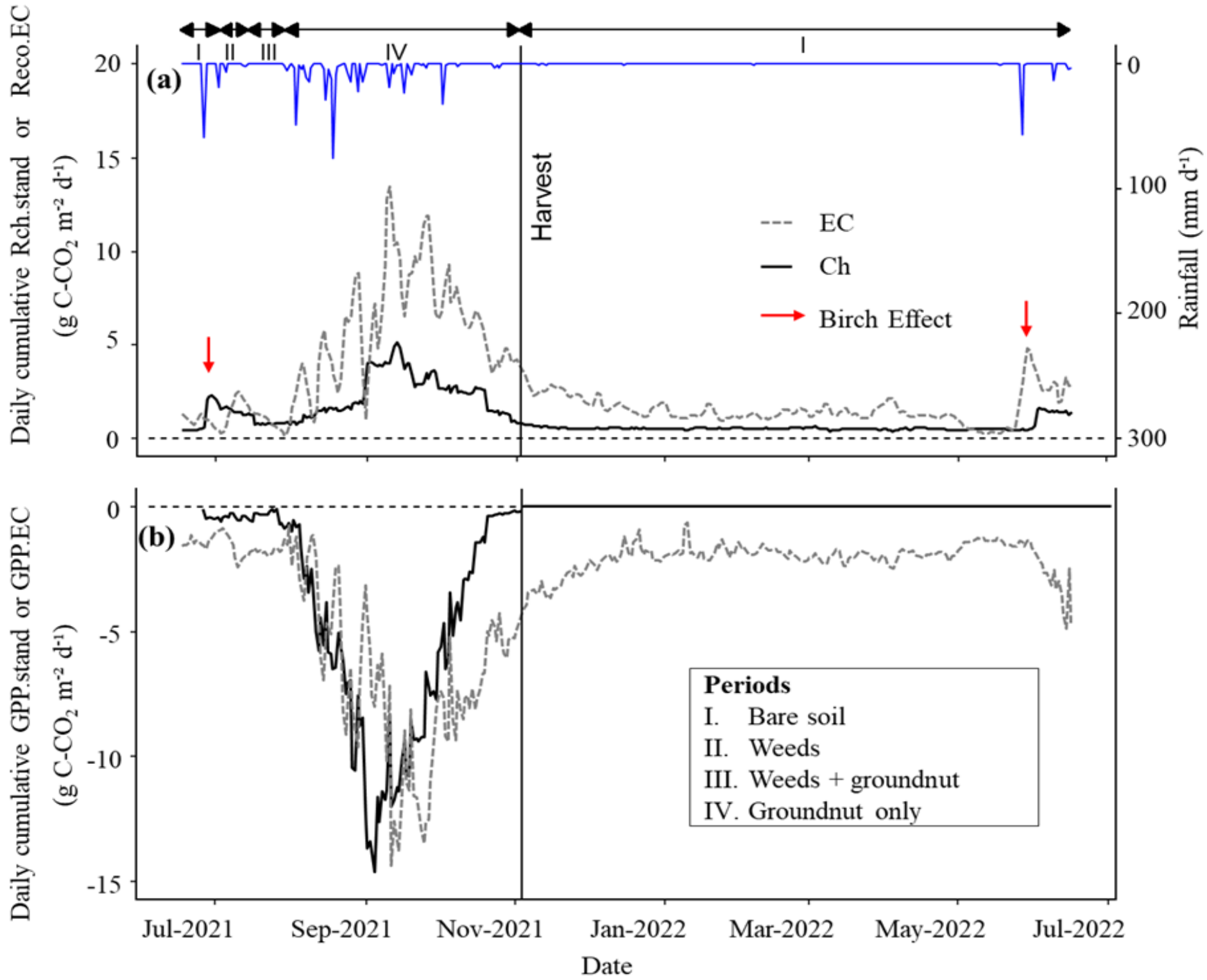

Fig 6: Comparing the seasonal dynamics of $CO_2$ fluxes between Eddy Covariance (EC) measurements and upscaled chamber measurements (ch.stand).

(a) represent the seasonal dynamics of soil + crop respiration (Rch.stand) and ecosystem respiration (Reco.EC) and (b) photosynthesis (GPP.stand and GPP.EC). The black and dashed grey lines show Ch and EC seasonal dynamics, respectively. The vertical black line indicates the harvest date of groundnuts inside the chambers. The blue line depicts the daily cumulative rainfall (mm $d^{-1}$), and the red arrow indicates the 'Birch' effect. Roman numerals (above the black arrows) refer to conditions prevailing inside the chambers: (I) bare soil, (II) weeds, (III) weeds + groundnuts, (IV) groundnuts.

Table 3: Comparison of gap-filled $CO_2$ fluxes between Eddy Covariance (EC) and upscaled chamber (Ch.stand) measurements, by season (rainy or dry).

| | Rainy season | | | | Dry season | | | |
|---|---|---|---|---|---|---|---|---|
| | Daily Mean ± SD | Min | Max | Mann-Whitney test | Daily Mean ± SD | Min | Max | Mann-Whitney test |
| (g C-$CO_2$ $m^{-2}$) | $.d^{-1}$ | $.d^{-1}$ | $.d^{-1}$ | | $.d^{-1}$ | $.d^{-1}$ | $.d^{-1}$ | |
| **Reco.EC or Rch.stand** | | | | | | | | |
| EC | 4.6 ± 3.2 | 0.2 | 13.5 | * | 1.2 ± 0.4 | 0.3 | 2.1 | * |
| Ch.stand | 2.0 ± 1.1 | 0.5 | 5.1 | | 0.5 ± 0.04 | 0.4 | 0.6 | |
| **GPP.EC or GPPch.stand** | | | | | | | | |
| EC | -5.1 ± 3.6 | -0.7 | -14.3 | * | -1.7 ± 0.3 | -0.6 | -2.4 | - |
| Ch.stand | -4.2 ± 4.3 | <-0.1 | -14.6 | | 0 | 0 | 0 | |

Mean, SD, Min, and Max represent the daily mean fluxes, standard deviation, minimum, and maximum values, respectively (g C- $CO_2$ $m^{-2}$ $d^{-1}$). The Asterisks (*) indicate the p-values from the Mann-Whitney test, used to assess differences in mean between EC and Ch. Positive values indicate $CO_2$ emissions, while negative values represent $CO_2$ uptake.

### *3.6. The contribution of F. albida to Reco and GPP*

During the dry season, the cumulative contribution of *F. albida* to ecosystem respiration (Ra tree) was 139.6 g C-$CO_2$ $m^{-2}$. This represent 12% of the total annual cumulative Reco, which was estimated at 1180.0 g C-$CO_2$ $m^{-2}$. The contribution of trees (GPP tree) to total annual GPP was -270.2 g C-$CO_2$ $m^{-2}$, equivalent to ~50% of the total annual cumulative GPP of the ecosystem (550 g C-$CO_2$ $m^{-2}$).

The ratio between these two components (Ra tree / GPP tree) in absolute terms was 0.52, reflecting a carbon use efficiency (CUE) of 0.48 (S3, Table S3.1).

### *3.7. Carbon budgets at the field-scale*

The upscaled chamber-based annual cumulative total respiration flux (Rch.stand) was estimated to be 4.1 $\pm$ 0.01 Mg C-$CO_2$ $ha^{-1}$ (Table 4). In comparison, the annual budget of Reco.EC was 10.0 $\pm$ 0.03 Mg C-$CO_2$ $ha^{-1}$ (Table 4), more than two times larger than Rch.stand.

The upscaled GPPch.stand reached an annual cumulative value of -5.5 $\pm$ 0.03 Mg C-$CO_2$ $ha^{-1}$, whereas the annual cumulative GPP.EC was -11.8 $\pm$ 0.03 Mg C-$CO_2$ $ha^{-1}$ (Table 4).

The annual net C budget, based on both methods, was estimated at -1.4 $\pm$ 0.02 Mg C-$CO_2$ $ha^{-1}$ for chambers ($FCO_2$ch.stand) and -1.8 $\pm$ 0.01 Mg C-$CO_2$ $ha^{-1}$ for Eddy Covariance (NEE.EC) (Table 4).

Table 4: Annual budget of $CO_2$ fluxes based on Eddy Covariance (EC) and upscaled chamber
methods (Ch.stand).

| | Annual sum (Mg C-$CO_2$ ha$^{-1}$) | Std error (Mg C-$CO_2$ ha$^{-1}$) |
|---|---|---|
| **Reco.EC or Rch.stand** | | |
| EC | 10.0 | 0.03 |
| Ch.stand | 4.1 | 0.01 |
| **GPP.EC or GPPch.stand** | | |
| EC | -11.8 | 0.03 |
| Ch.stand | -5.5 | 0.03 |
| **NEE.EC or F$CO_2$ch.stand** | | |
| EC | -1.8 | 0.01 |
| Ch.stand | -1.4 | 0.02 |

Annual sum corresponds to the annual cumulative fluxes for full year measurements (Mg C-$CO_2$ ha$^{-1}$). EC
refers to fluxes measured by the Eddy Covariance method, and Ch refers to the fluxes measured by
chambers, which are then upscaled to the whole field. Rch.stand represents the chamber respiration, while
Reco.EC denotes the ecosystem respiration according to the EC method. GPP.EC and GPPch.stand are the
gross primary production or photosynthesis flux, measured by EC and Ch methods, respectively. NEE.EC
and F$CO_2$ch.stand represent the net ecosystem exchange for EC and Ch, respectively. The associated
standard error is denoted as Std error (Mg C-$CO_2$ ha$^{-1}$). Positive values indicate $CO_2$ emissions, while
negative values represent $CO_2$ uptake.

# 4. Discussion

## 4.1. *Seasonality and drivers of chamber-based $CO_2$ fluxes*

In our agroforestry context, seasonal variability in $CO_2$ fluxes closely followed rainfall dynamics, peaking during the wet season and declining sharply in the dry season, consistent with soil moisture depletion and crop senescence. This pattern is typical of semi-arid ecosystems (Ago et al., 2016a; Brümmer et al., 2008; Guillen-Cruz et al., 2023; Macharia et al., 2020; Mosongo et al., 2022; Wieckowski et al., 2024).

Respiration and photosynthesis were primarily driven by soil moisture and LAI, reflecting the system's sensitivity to water availability and crop dynamics. Soil moisture enhanced both processes by stimulating microbial activity and supporting plant growth (Borken et al., 2002; Conant et al., 2004; Merbold et al., 2009; Yu et al., 2020; Zhao et al., 2016). The stronger correlation between soil moisture and respiration under F. albida canopy (Sh: $r = 0.78$) compared to full sun (FS: $r = 0.51$) suggests greater microbial sensitivity to moisture beneath trees. This likely reflects enhanced substrate availability, resulting in stronger post-rainfall respiration pulses (Meisner et al., 2015) and supporting the 'fertile island' effect, where trees improve local soil conditions (Eldridge et al., 2024). Photosynthetic capacity also responded to soil moisture, as shown by positive correlations with LAI and key physiological traits such as light use efficiency ($\alpha$) and maximum $CO_2$ uptake rate ($\beta$) (Gonsamo et al., 2019; Qiu et al., 2023; Zhang et al., 2024).

In contrast, the influence of soil temperature ($T_{soil}$) on respiration was weakly negative in both FS and Sh, indicating a thermal threshold beyond which respiration is suppressed—estimated at 32 $\pm$ 1.5 °C in FS and 29.5 $\pm$ 1.9 °C in Sh (S2, Fig. S2.6, a and b), similar to findings in Eastern Ghana (Owusu et al., 2024). This inhibition likely results from decreased enzymatic and microbial activity under combined heat and water stress (Liu et al., 2018; Richardson et al., 2012). In semi-arid regions, soil respiration often becomes decoupled from temperature due to seasonal moisture constraints (Jia et al., 2020; Tucker & Reed, 2016; Warren, 2014), with microbial activity limited during dry periods despite favourable temperatures. This decoupling helps explain the weak or absent correlation between $T_{soil}$ and soil moisture (S2, Fig. S2.5, b), particularly under Sh ($r = -0.28$). Management practices such as organic inputs can also modulate these dynamics, adding further variability to soil respiration responses (Meena et al., 2020; Oyonarte et al., 2012; Rong et al., 2015; Xue & Tang, 2018).

## 4.2. *Magnitude of chamber-based total $CO_2$ respiration fluxes*

Mean total soil respiration values were consistent with those reported in other low-input agricultural systems across sub-Saharan Africa (Mapanda et al., 2010; Pelster et al., 2017; Rosenstock et al., 2016). In full sun (FS), the mean respiration ($1.0 \pm 0.9$ g C-$CO_2$ $m^{-2}$ $d^{-1}$) closely

matched values measured by Wachiye et al. (2020) in a semi-arid Kenyan field at 1158 m altitude (1.1 ± 0.1 g C-$CO_2$ $m^{-2}$ $d^{-1}$). This similarity likely reflects comparable environmental conditions, including moderate rainfall (~550 mm $yr^{-1}$) and low soil organic carbon and nitrogen contents (<1%) in the 0–20 cm layer of sandy soil. In contrast, respiration under *F. albida* canopy (Sh: 1.6 ± 1.1 g C-$CO_2$ $m^{-2}$ $d^{-1}$) was higher, likely due to additional autotrophic respiration from tree roots and greater organic inputs beneath the canopy. Nonetheless, this flux remains close to values observed in low-input sorghum fields on sandy loam soils in eastern Ghana (1.7 ± 1.1 g C-$CO_2$ $m^{-2}$ $d^{-1}$), despite higher rainfall (950–1000 mm $yr^{-1}$) in that region (Owusu et al., 2024).

Cumulative annual respiration fluxes fell within the range reported for Sahelian croplands (250–450 g C-$CO_2$ $m^{-2}$) (Brümmer et al., 2009) and other sub-Saharan African agricultural systems (Kim et al., 2016). The cumulative flux under tree cover is similar to that measured in cassava fields in eastern Tanzania (440 g C-$CO_2$ $m^{-2}$ $yr^{-1}$), despite the latter receiving higher rainfall (~1115 mm $yr^{-1}$) (Rosenstock et al., 2016). This convergence may stem from comparable soil fertility constraints, with low soil organic carbon (1–1.7%) and nitrogen contents (<0.5%). In contrast, the slightly lower cumulative flux in FS may reflect less favourable microclimatic conditions—such as elevated soil temperatures and increased aridity away from tree cover—limiting microbial activity (see Section 4.1).

Across sub-Saharan Africa, soil respiration fluxes based on static chamber measurements show high spatial variability, largely shaped by climate and land use. For example, Owusu et al. (2024) found higher respiration in woodlands (3.8 ± 0.8 g C-$CO_2$ $m^{-2}$ $d^{-1}$) and grazed areas (2.7 ± 1.7) than in croplands (1.7 ± 1.1) in humid eastern Ghana. This gradient was linked to differences in soil moisture and organic matter. Similarly, Rosenstock et al. (2016) reported much higher fluxes in highland pastures in Kenya (3.8–4.4 g C-$CO_2$ $m^{-2}$ $d^{-1}$) compared to cultivated fields in eastern Tanzania (1.2 ± 0.2), highlighting the role of vegetation cover and soil fertility.

### *4.3. Effect of trees on chamber-based soil respiration and photosynthesis*

A notable increase in respiration and photosynthesis fluxes was observed under *F. albida* trees (Sh) compared at a distance from trees (FS). This increase may indicate the potential role of *F. albida* in modulating $CO_2$ exchange dynamics (Rch and GPPch) within this agro-silvo-pastoral system. These results are consistent with preliminary findings from similar environments (Duthoit et al., 2020).

Numerous studies have investigated the effect of tree species on greenhouse gas fluxes, particularly $CO_2$, revealing significant variations across different ecological contexts (Bréchet et al., 2021, 2025; Klaus et al., 2024; Mazza et al., 2021; Ramesh et al., 2013; Rheault et al., 2024). However, the underlying mechanisms by which trees influence these dynamics are not yet fully understood.

In general, agroforestry systems have been well-documented for their ability to provide a range of ecosystem services (e.g., Assefa et al., 2024; Bado et al., 2021; Kuyah et al., 2019; Rolo et al., 2023). Specifically, *Faidherbia*-based agroforestry systems may play a crucial role in regulating $CO_2$ exchanges between the soil and atmosphere. *F. albida*-based agroforestry systems are recognized for enhancing both soil organic and mineral fertility (Bayala et al., 2020; Dilla et al., 2019; Sileshi, 2016; Sileshi et al., 2020; Stephen et al., 2020), mainly through litter accumulation and direct inputs from livestock excreta under their canopies. Additionally, the extensive roots system of *F. albida* trees helps concentrate mineral nutrients, contributing to the formation of a 'fertile island' effect under the trees (Siegwart et al., 2022; Eldridge et al., 2024). Moreover, *F. albida* improve water infiltration (Diongue et al., 2023; Faye et al., 2020; Sarr et al., 2023), enhance soil moisture retention (Clermont-Dauphin et al., 2023) and contribute to reduced soil temperatures (de Carvalho et al., 2021; Lopes et al., 2024; Sida et al., 2018). These changes foster a more favourable environment for soil microbial activity and crop development (Diack et al., 2024; Diene et al., 2024; Leroux et al., 2020; Roupsard et al., 2020) under the trees compared to open areas. Consequently, this likely explains the stronger effect of soil moisture and the leaf area index of groundnuts on Rch under the trees, resulting in higher total respiration (Table 2). For photosynthesis, the effect of these parameters was similar in both FS and Sh (Table 2). However, the significantly higher intensity of GPPch under Sh can be explained by greater light use efficiency ($\alpha$) and a higher maximum $CO_2$ uptake rate at light saturation ($\beta$) in this shaded environment. In agroforestry systems, light use efficiency can at least partially mitigate the reduction in photosynthetically active radiation under tree canopies (Charbonnier et al., 2017).

Similar results have been observed in different climatic conditions and ecosystems. Gomes et al. (2016) investigated soil respiration using mobile chambers (LI-8100-102 model) under trees in coffee-based agroforestry (AF) systems and in open areas (FS) in Minas Gerais, Brazil. These studies were conducted with agroecological management practices, such as weeding, intercropping maize between coffee rows, and mulching. The AF systems exhibited lower air and soil temperatures (at 5 and 10 cm depth) and higher air and soil humidity compared to FS (Gomes et al., 2016). These authors observed greater spatial variability in soil respiration in AF (34.1%) compared to FS (24.2%). This variability was mainly linked with fluctuations in labile carbon and total nitrogen, reflecting more favourable soil microclimate for microbial activity in AF. In contrast, soil temperature (10 cm depth) accounted for most of the variability observed in FS, where the absence of tree canopy resulted in high soil temperatures and low soil moisture (Gomes et al., 2016). Likewise, Haren et al. (2010) reported 38% higher soil respiration near large trees (DBH > 35 cm) in clay-rich Amazonian forests compared to open sites. Interestingly, the magnitude of $CO_2$ fluxes was independent of tree species, indicating that canopy effects may outweigh species-specific traits in some contexts. In our study, *F. albida*'s influence on $CO_2$ fluxes

aligns with this general pattern observed in tropical agroforestry. However, the mechanisms linking individual tree species to microbial and physicochemical drivers of $CO_2$ dynamics remain insufficiently understood and warrant further investigation (Jevon et al., 2023).

### *4.4. Birch Effect*

A rapid increase in soil respiration was observed following the first rainfall events, particularly under *F. albida*. This phenomenon can be attributed to the lower bulk density of the soil under the trees (Clermont-Dauphin et al., 2023; Siegwart et al., 2023), which potentially lead to $CO_2$ accumulation during the dry season due to higher soil organic matter (SOM) (Siegwart et al., 2023). Additionally, the sensitivity of microbial communities to subtle variations in soil moisture, compounded by the tree effect, may further explain this phenomenon, as outlined in Sections 4.1 and 4.3. This phenomenon, known as the 'Birch effect' (Birch, 1958), has been reported across various semi-arid ecosystems in sub-Saharan Africa (Ago et al., 2016b; Fan et al., 2015; Wieckowski et al., 2024), as well as other semi-arid ecosystems globally (Roby et al., 2022; Yan et al., 2014; Yu et al., 2020). In these contexts, the 'Birch effect' may result from the displacement of soil gas phases by the piston effect generated during water infiltration (Singh et al., 2023). Furthermore, microbial communities in semi-arid environments adopt osmoregulatory mechanisms to withstand water deficit (Warren, 2014), which is particularly pronounced during the dry season. This phenomenon reduces soil microbial metabolism (Schimel et al., 2007). Upon rapid soil rewetting, especially after prolonged dry periods, soil microbial metabolism process is swiftly reactivated, leading to a transient pulse in respiration and a $CO_2$ release (Barnard et al., 2020; Kim et al., 2012; Manzoni et al., 2020; Vargas et al., 2018). Isotopic signatures of soil respiration provide evidence supporting the hypothesis that these pulses result from the rapid mineralisation of necromass or osmolytes excreted by microorganisms under drought stress (Schimel et al., 2007; Unger et al., 2010). Additional factors may amplify the 'Birch effect'. For instance, drying-rewetting cycles can induce physical disruption of soil aggregates, enhance oxygen penetration and thereby expose previously protected organic matter to microbial decomposition (Rabbi et al., 2024). This increases substrate availability and subsequently boosts soil respiration fluxes.

The magnitude of the 'Birch effect' is modulated by the severity and duration of drought. Thus, at our study site, given the 8- to 9-month-long dry season, the 'Birch effect' is particularly intense. Indeed, extended drought periods promote greater accumulation of microbial necromass and intensify hypo-osmotic stress responses upon rewetting (Singh et al., 2023).

### *4.5. Comparing EC and chamber-based methods*

Results revealed high seasonal variability, with higher values during the rainy season compared to the dry season. This seasonal pattern aligns with findings from studies in the Sahel using the EC method for $CO_2$ flux measurements (Brümmer et al., 2008; Tagesson et al., 2015; Agbohessou et al., 2023, Wieckowski et al., 2024). Comparable patterns have been also documented at the ecosystem scale in other semi-arid environments (Ago et al., 2014; Archibald et al., 2009; Ardö et al., 2008; Jia et al., 2020; Quansah et al., 2015; Williams et al., 2009; Zhang, Bi, et al., 2024).

Several comparative studies between chamber and EC methods have reported both congruent and divergent $CO_2$ flux estimates (Bastviken et al., 2022; Poyda et al., 2017; Riederer et al., 2014; J. Tang et al., 2008; Wang et al., 2010). In the present study, ecosystem respiration fluxes during the rainy season exhibited notable discrepancies measurements between EC (Reco.EC) and upscaled chamber-based (Rch.stand). This is attributable to differences in the flux components captured by each method. Specifically, Reco.EC included respiration from below- and above-ground tree parts, crops (groundnuts and cowpeas), weeds, and soil, whereas Rch.stand accounted only respiration from below-ground tree, groundnut crop, and soil. Therefore, as expected, Reco.EC (4.6 ± 3.2 g C-$CO_2$ $m^{-2}$ $d^{-1}$) were significantly higher than Rch.stand (2.0 ± 1.1 g C-$CO_2$ $m^{-2}$ $d^{-1}$).

For chamber-based GPP measurements, values were standardised (GPP-stand) by the field's leaf area index (LAI.field). This allowed it to improve comparability with GPP.EC when trees were leafless in the rainy season. In both cases, GPP accounted only for crops (groundnut and cowpea) and weeds, as trees were non-photosynthetic in the rainy season. Despite this standardisation, GPP.EC values (-5.1 ± 3.6 g C-$CO_2$ $m^{-2}$ $d^{-1}$) were significantly higher than GPPch.stand values (-4.2 ± 4.3 g C-$CO_2$ $m^{-2}$ $d^{-1}$). However, the divergence did not occur on the peak of GPP (which was very similar in both methods), but from the onset of groundnut senescence, when weeds became the dominant photosynthetic contributors. Thus, during the groundnut growth season, with leafless *F. albida* trees and almost no weeds, GPP measurements from EC and chambers generate closely comparable results. Therefore, this provides an initial form of cross-validation between the two methods. It is important to note that the EC method integrates $CO_2$ fluxes over a larger spatial scale, encompassing all ecosystem components (Baldocchi, 2003), while the chamber method captures fluxes on a smaller scale (i.e., at the 0.25 $m^2$ scale). This scale disparity can introduce uncertainties when upscaling chamber-based fluxes to the field, as vegetation composition within chambers does not represent the EC footprint's average vegetation. This makes upscaling chamber-based measurements challenging. Nevertheless, the standardisation we applied on chamber photosynthesis by LAI has been relatively successful.

During the dry season, Reco.EC included respiration from below- and above-ground tree parts (with leaves) and bare soil, while Rch.stand measured only below-ground tree and bare soil

respiration. Consequently, the difference between Reco.EC and Rch.stand was solely attributable to above-ground tree respiration (Ra tree above-ground). In terms of GPP, chamber measurements were nil, whereas GPP.EC reflected only GPP trees.

The transition period, characterised by groundnut senescence, tree leaf regrowth, and weed proliferation, introduced further complexity, amplifying method-specific discrepancies. Rch.stand measurements facilitated the estimation of tree contribution to Reco.EC (Ra tree) and the verification of the consistency for EC results in terms of carbon use efficiency (CUE), estimated here at 0.48. This value indicates that nearly 50% of the carbon captured by trees is allocated to biomass. The CUE estimate here is well comparable to the global average across diverse ecosystems, climates, and management practices (0.49 $\pm$ 0.14) (Tang et al., 2019). Similar CUE values have been reported for semi-arid grasslands (0.46 $\pm$ 0.10), but our value is notably lower than those documented for wetlands (0.61 $\pm$ 0.13) (Tang et al., 2019). Overall, these findings reinforce the plausibility of our assumptions regarding the compartment's contributions to Reco.EC and Rch.stand, thereby providing a second cross-validation of the EC-Ch comparison. However, despite a frequently assumed CUE of 0.5 in models, global estimates span a broad range (0.20 to 0.82), depending on ecosystem type and management practices (DeLucia et al., 2007; Tang et al., 2019). This underscores the importance of refining carbon flux models to better represent the biophysical processes governing $CO_2$ exchange in semi-arid agroforestry systems. The combined use of EC and chamber methodologies offers a comprehensive perspective on ecosystem-scale $CO_2$ flux dynamics, advancing understanding of carbon cycling in these environments.

### *4.6. Net carbon exchange budget*

The annual net carbon (C) exchange budget was quantified at -1.4 $\pm$ 0.02 Mg C-$CO_2$ $ha^{-1}$ with the chamber method and -1.8 $\pm$ 0.01 Mg C-$CO_2$ $ha^{-1}$ by the Eddy Covariance (EC), indicating that the studied agro-silvo-pastoral system functions as a net carbon sink. These findings corroborate the system's potential role in mitigating greenhouse gas emissions, consistent with previous observations in semi-arid ecosystems (Rahimi et al., 2021; Tagesson et al., 2015; Agbohessou et al., 2023, Wieckowski et al., 2024).

The estimated net C exchange budget is close to the reported mean for Sahelian ecosystems (-1.6 $\pm$ 0.5 Mg C-$CO_2$ $ha^{-1}$; Tagesson et al., 2016). The EC-based net C exchange budget (-1.8 $\pm$ 0.01 Mg C-$CO_2$ $ha^{-1}$) is also similar to the value of -1.9 $\pm$ 0.4 Mg C-$CO_2$ $ha^{-1}$ reported for semi-arid savannas of northeastern Benin, despite higher annual rainfall (1495 mm; Ago et al., 2016b). Furthermore, our EC estimate is close to the average net C exchange reported for West African terrestrial ecosystems (-2.0 $\pm$ 1.5 Mg C-$CO_2$ $ha^{-1}$; Ago et al., 2016a).

However, estimates from Tagesson et al. (2015) (-2.7 ± 0.07 Mg C-$CO_2$ $ha^{-1}$) for a semi-arid savannah in Dahra, Senegal, located between the 300 mm and 400 mm isohyets, were comparatively higher. This is potentially attributable to specific characteristics of that specific savannah site, such as herbaceous vegetation cover during the rainy season, the presence of evergreen trees, and land management practices linked to pastoral livestock activities (Tagesson et al., 2016).

The net C exchange estimates presented in this study are, in fact, apparent fluxes, given that they exclude organic matter (OM) imports and, more critically, exports, introducing uncertainties. Notably, the export of crop residues and direct inputs from animal excreta —particularly significant in 'bush fields' during the dry season — were not accounted for. In our case of 'bush field', crop residues are exported to feed livestock, while livestock faeces are collected for use as fuel or manure in 'home fields'. Such practices may lead to a significant soil organic carbon stocks depletion (Malou et al, 2021), potentially diminishing the net C budget (-1.4 ± 0.02 Mg C-$CO_2$ $ha^{-1}$) over time and shifting the system closer to carbon neutrality (Assouma et al., 2019).

These results should be contextualized within the broader framework of climate change and semi-arid ecosystem management. Although agro-silvo-pastoral systems can function as annual carbon sinks, they remain highly sensitive to interannual rainfall variability and escalating anthropogenic pressures. Sustainable management practices, particularly regarding crop residue exports, are essential for maintaining soil mineral fertility and preserving the system's capacity to act as a carbon sink, thereby contributing to climate change mitigation.

### *4.7. Limitations of the study*

This study benefited from the inverse phenology of F. albida, allowing for direct comparison between chamber-based GPP (GPPch.stand) and ecosystem-level GPP (GPP.EC) during the leafless period of the trees. However, the system's spatial heterogeneity —common in agroforestry— posed challenges for accurately partitioning $CO_2$ fluxes among trees, crops, and soil. A key limitation was the development of weeds during the late rainy season, which complicated the attribution of fluxes, particularly during the transitional period. Additionally, while GPPch was successfully standardised by LAI for upscaling, this was not feasible for respiration. Respiration integrates both autotrophic and heterotrophic components, which respond to different drivers and are not directly linked to LAI, limiting the precision of upscaled Rch.

Future improvements should aim to separately quantify respiration sources —tree roots, crops, and microbial (heterotrophic) respiration— and account explicitly for the weed layer, to refine flux partitioning in such complex agroforestry systems.

## Conclusion

This study demonstrates the successful application of automated static chambers to quantify $CO_2$ fluxes in a Sahelian agroforestry system dominated by *F. albida*. The continuous, high-frequency measurements captured key seasonal dynamics and short-lived events (e.g., Birch effect), providing a more accurate assessment of carbon exchange than traditional intermittent sampling. By integrating crop and soil components and applying dynamic partitioning models, the study quantified both respiration and photosynthesis fluxes at fine temporal resolution. The results revealed a clear 'fertile island' effect under tree canopies, with higher respiration and photosynthetic activity, and highlighted the significant contribution of F. albida trees to annual carbon uptake.

The consistency between chamber- and eddy covariance-based estimates reinforces the robustness of the methodology. Overall, this work underscores the role of F. albida-based agroforestry systems as effective carbon sinks in semi-arid environments, offering valuable insights for carbon accounting and sustainable land management in the Sahel.

**Acknowledgments**

This research was financially supported by the CaSSECS project (Carbon Sequestration and Greenhouse Gas Emissions in (Agro) Silvopastoral Systems of the CILS-Sahel States (FOOD/2019/410-169), within the framework of the European Union's initiative 'Development of Smart Innovation through Agricultural Research' (DeSIRA-UE-EuropAID). We extend our sincere gratitude to the coordination team of the CaSSECS project, the ''Laboratoire Mixte International Intensification Écologique des Sols Cultivés en Afrique de l'Ouest'' (LMI IESOL) of the of the French National Institute for Development (IRD) in Dakar (Senegal), as well as to the Faidherbia-Flux platform (https://lped.info/wikiObsSN/?Faidherbia-Flux), its partners, and affiliated projects PEPR FairCarbonN/PC3-RIFT, EU-H2020 |SUSTAIN-SAHEL (Grant N° 861974)] and EU-HORIZON EUROPE [GALILEO (Grant N° 101181623) ]. Our deepest appreciation goes to Ibou Diouf, the observer at our experimental site. Tagesson also acknowledged funding from Formas (Dnr 2021-00644).

**Author contribution: CRediT**

**Seydina Mohamad BA**: Conducting in situ experiments, collecting and processing data, writing-original draft, review and editing. **Olivier Roupsard**: Designing experimental apparatus and methodology, writing, review and editing. **Lydie Chapuis-Lardy**: Designing methodology, writing, review and editing. **Yélognissè Agbohessou**: Processing data, review and editing. **Fred Bouvery**: Designing chambers and connection to the instrument, review and editing. **Maxime Duthoit**: Designing experimental set and methodology, review and editing. **Aleksander Wieckowski**: Review and editing. **Mohamed Habibou Assouma**: Review and editing. **Espoir Gaglo**: Processing data, review and editing. **Claire Delon**: review and editing. **Torbern Tagesson**: Designing methodology, review, and editing. **Bienvenu Sambou**: Review and editing. **Dominique Serça**: Designing methodology, writing, review and editing.

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
