# Peer review of "Drivers and CO2 flux budgets in a Sahelian *Faidherbia albida* agro-silvo-pastoral parkland"

_EGUsphere, 2025_

## Referee Comment (RC2)

**Review comments on "Drivers and CO$_2$ flux budgets in a Sahelian Faidherbia albida agro-silvo-pastoral parkland: Insights from continuous high-frequency soil chamber measurements and Eddy Covariance" – Ba et al. (EGUsphere)**

The research of Ba et al. focuses on CO$_2$ flux dynamics of an increasingly popular agro-forestry land use in Africa, featuring *Faidherbia albida* trees, *groundnut* plants and livestock grazing. All chapters are neatly written which gives the reader a full and clear picture of the research that has been done and the results that were collected. Overall, the research seems to be conducted well and features interesting findings. Most of all, the authors have quantified GPP and Rh of various ecosystem elements and showed how these elements fit within the bigger picture of the complete ecosystem. This increases our general understanding of these systems, which is needed to enable improvements of land-use in the longer term. Moreover, the results that are presented can be of high value for ecosystem and/or climate models as the (Sahel) region seems to be, as the authors state, particularly underrepresented in global carbon flux research. Nevertheless, I do have three major concerns or questions that I would like to mention below.

First, I am concerned about the methodology to partition and extrapolate CO2 fluxes discussed in section 2.3.3. The authors discuss the assumptions on which the Arrhenius-type function from Lloyd & Taylor (1994) relies that was used for extrapolating ecosystem respiration. The first assumption features an exponential response between soil temperature and respiration. However, the authors also describe that high (soil) temperatures suppress daily respiration (discussion section 4.1). This is attributed to a decreased microbial activity which suppressed soil respiration and has been described more often in literature. The authors not only found that respiration is suppressed at higher temperatures, but they also even mention a (weak) negative correlation between respiration and soil temperatures (section 3.4). Figure S2.6 shows, besides a suppression of Rch due to high temperatures, that there does not seem to be a clear exponential temperature relation. This raises questions about the validity of the assumption on which the partitioning, extrapolation and gap-filling of CO2 fluxes were based. The authors show that the nocturnal respiration can be modelled quite well in Fig. 3, but how does this translate to daytime when the temperatures are higher? Given the observed negative correlation, the authors should justify their approach. If the model is inappropriate under high temperatures, a different approach might be needed.

Second, I have a question about non-linearity which could affect chamber flux results. When working with the chambers the authors noted fogging and decided to shorten the flux-analysis from 15 to 5 minutes during the groundnut growing season. Other causes may still lead to a non-linear measurement of CO2 concentrations after chamber closure. For example, a high plant uptake of CO2 could diminish CO2 concentrations

substantially, eventually slowing down plant uptake. When a flux is calculated using a fitting period that is too long, the slope of the CO2 uptake will be lower than the initial slope, misrepresenting the actual initial CO2 uptake and affecting the total CO2 balance. How did the authors make sure this non-linearity was minimized during the flux calculations? Did some 5-minute flux measurements turn out to be non-linear? If so, how were these cases handled? Were any non-linear fluxes excluded by filtering fluxes that had a R2 < 0.8? Would that be the right choice?

Third, the authors present an annual carbon budget of the ecosystem that was measured but did not include harvest and livestock manure C-terms. Even though the authors clearly mention and discuss this problem in the methods and discussion sections, I have my doubts about the usage of the term *carbon budget*. When the livestock was not fed externally, and manure is not exported from the system, we could assume that the presence and grazing would have a marginal impact on the carbon budget. However, in the discussion it is mentioned that faeces are collected from the field. Furthermore, biomass harvest C-export normally represents a substantial term within a carbon budget of an agricultural system. I do understand that a carbon budget is a valuable result. However, ignoring these C-terms and then comparing the carbon budget to literature seems incorrect and may lead to misleading comparisons. Would it be possible to roughly estimate the missing components to construct an actual carbon budget? The estimates could feature substantial errors that can be propagated. Such an approach may provide a more complete carbon budget and facilitate a fair comparison with other studies.

Overall, the manuscript is valuable and presents interesting findings. Addressing the issues outlined above would strengthen it further. Below, I list several other minor issues.

**Minor comments:**

*Highlights*

The highlights include abbreviations (Sh, FS) that are unknown to readers.

*Introduction*

Line 99. Please check the usage of present time.

Line 225. Please remove the repetition.

*Results*

Line 443. Table 1 results for daily FCO2 are negative, while numbers here appear positive.

Table 2. It is a choice to not denote non-significant correlations. However, a p-value of 0.05 is arbitrary. There might be different visions on this matter, but I would not 'hide' non-significant correlations and show each p-value (or p-value category).

Table 4. How was the std error that is shown calculated?

*Discussion*

Section 4.5. Sometimes it is hard to follow which periods are being discussed. In general, it could help to specifically mention the months that are being discussed.

Line 704. The authors mention that chamber and EC GPP measurements agree closely. I do agree that this is the case in August, but after the beginning of September the two seem to start deviating remarkably. As mentioned above, please clarify which months are under discussion.

Section 4.6. Please see third point above.

*Conclusion*

Line 794. Since the actual carbon balance is unknown, it cannot be stated that the agroforestry systems that were studied are 'effective carbon sinks'.

---

## Author Comment (AC2)

**Title:** *Drivers and CO₂ flux budgets in a Sahelian Faidherbia albida agro-silvo-pastoral parkland: Insights from continuous high-frequency soil chamber measurements and Eddy Covariance.*

**Author (s):** Seydina Mohamad Ba et al.

**MS type:** Original research article

**Manuscript No.:** egusphere-2025-2660, submitted to Soil
* * *
Dear Riccardo Picone,

We sincerely thank you for your careful and thorough review of our manuscript, as well as for the positive and encouraging tone of your report. Your comments and suggestions have been extremely valuable and have significantly contributed to improving both the scientific rigour and the readability of the paper.

We have addressed all of your points in the revised version. For clarity, please note that the line numbers indicated in the responses are provisional, as additional revisions may result in changes to the manuscript's line numbering. We will ensure consistency in the final revised manuscript.

Below is a detailed point-by-point response to each of your comments. For ease of reviewing, the reviewer's original comments are reproduced in *black italics*, our responses are in black regular text, and the corresponding changes made to the manuscript are indicated in blue and bold.

With best regards,

On behalf of all the co-authors,

Seydina Mohamad BA,
PhD candidate in the EU-funded CASSECS project
IESOL, Centre IRD-ISRA, 18524, Dakar; Senegal
Email: seydina.ba@ird.fr
and
Olivier Roupsard
Researcher
CIRAD, UMR Eco&Sols, Dakar; Sénégal
Eco&Sols, Univ Montpellier, CIRAD, INRAE, Institut Agro, IRD, Montpellier; France
IESOL, Centre IRD-ISRA de Bel Air, 18524, Dakar; Senegal
Email: olivier.roupsard@cirad.fr

**Authors' response to reviewer RC1**

**Reviewer's report:**

*This work reports a comparison of carbon fluxes between different zones of an agroforestry system during one whole year assessed with two different methodologies. A comparison of the two methodologies was also done. The topic is therefore highly relevant for the journal. The abstract does a good work in framing the context of the study and the relevance of the findings. The introduction effectively presents the topic and its importance, by highlighting key knowledge gaps that will be addressed by the study. Anyway, I would suggest including a brief description of the Eddy Covariance method in this section. In my opinion, the main problem of the paper is that clear starting hypotheses that have driven the work were not stated. This should be addressed. Methodologies appear to be consistent and appropriately described, and all the reported methods have the appropriate bibliographic reference. I would only suggest some minor integrations to the experimental design description. Results are correctly reported in all the necessary detail. The discussion does a very good job in comparing the results to other studies, hypothesizing mechanisms driving the findings, and highlighting limitations of the study. Conclusions realistically summarize the key discoveries. All supplementary materials are relevant and correctly reported. The authors are requested to carefully proofread the "references" section because some journal names are not correctly abbreviated. Based on these considerations, I would recommend minor revisions to be applied to the manuscript before it can be accepted for publication.*

*Hereafter follow the specific comments I made on the text. Text between quotation marks indicates citations from the manuscript. When multiple lines are indicated, the comments refer either to a full sentence or to a meaningful part of it*

—---------------------------------------------------------------------------------------------------------------

**Comment #1**

*L. 43 Please report the full name of the species when it is first mentioned in the abstract.*

**Response #1**

The full name of the species was mentioned

Line 43: "*Faidherbia albida*" (Revised version)

**Comment #2**

L. 57 I would suggest deleting the phrase "of trees".

**Response #2**

The correction was done.

Line 57: "fertile island effect" (Revised version)
* * *
**Comment #3**

*L. 64 I would suggest briefly describing the methodology used for this technique.*

**Response #3**

The methodology used for EC technique is described as follows:

"The EC technique quantifies $CO_2$ exchanges between ecosystems and the atmosphere by correlating fluctuations in vertical wind velocity with simultaneous variations in $CO_2$ concentrations, providing a direct and non-invasive estimate of $CO_2$ fluxes (Baldocchi, 2003)."

Line 66 to 69 (Revised version)
* * *
**Comment #4**

*L. 95 "upscaling" Undertsanding/comprehension?*

**Response #4**

Thank you for pointing this out.

The main idea, here, is to partition the ecosystem fluxes by compartments (soil and trees).

The corrected sentence becomes:

"When combined with EC, this dual-method approach strengthens source attribution and improves the partitioning of fluxes across complex agroforestry landscapes".

Line 97 to 99 (Revised version)
* * *
**Comment #5**

*L. 99 The hypotheses that have driven the study are not stated.*

**Response #5**

Thank you very much for this comment, which further improves the quality of the paper.

We have therefore added the following assumptions:

(1) Rch and GPPch are higher under the canopy of F. albida than in full sun, (2) soil moisture is the main environmental factor directly controlling both Rch and GPPch, (3) when extrapolated to the field scale, the chamber-based method provides seasonal dynamics of respiration and photosynthesis fluxes comparable to those derived from EC technique.
Line 108 to 112 (Revised version)
* * *
**Comment #6**

*L. 139 At which distance from the trees were the chambers installed?*

**Response #6**

This was already mentioned in non-revised version line 139 (...at least 20 m from trees).
Now in line 148 (Revised version)
* * *
**Comment #7**

*L. 147 "half-hour flux measurements" Does this mean that the measurement was repeated every 30 mins in each chamber? If so, I suggest being more clear on this.*

**Response #7**

This means that a full flux measurement sequence (chamber closure, data acquisition, chamber opening, and purging) takes 30 minutes, before moving on to the next chamber, and so on.
* * *
**Comment #8**

*L. 160 "indicated" Indicating?*

**Response #8**

The correction has been made.
Line 169 : "indicating" (Revised version)
* * *
**Comment #9**

*L. 166, 168 "NDVI", "LAI" Please report the full name.*

**Response #9**

The full name for NDVI (normalised difference vegetation index) and LAI (leaf area index) was reported.
Lines 175 and 178 (Revised version):
* * *
**Comment #10**

*L. 172-174 How often were VWC and Tsoil measurements repeated?*

**Response #10**

"All at 5-min intervals", this measurement frequency for VWC and Tsoil was already indicated in non-revised version line 174.

Line 184 (Revised version).

**Comment #11**

*L.223-227 These two sentences are a repetition.*

**Response #11**

The repetition has been removed.

Line 231 to 233 (Revised version)

**Comment #12**

*L. 344 Inside "the" chambers.*

**Response #12**

The missing article ''the" has been added.

Line 368 (Revised version)

**Comment #13**

*L.351 "references" Reference*

**Response #13**

The correction has been made.

Line 376 (revised version)

**Comment #14**

*L.s 387-388 How do you account for a standard error of the same entity of the measurement itself?*

**Response #14**

Here, it refers to the mean $\pm$ standard deviation, not the standard error. I have added this clarification in the text.

The sentence now reads: "FS showing (mean $\pm$ standard deviation) $0.9 \pm 0.9$ μmol $CO_2$ $m^{-2}$ $s^{-1}$ (modeled) and $1.3 \pm 1.2$ μmol $CO_2$ $m^{-2}$ $s^{-1}$ (measured)".

Line 412 to 413 (Revised version).
* * *
**Comment #15**

*L. 483 "GPPshowed" A space is needed here.*

**Response #15**

Corrected.

Line 483 (Revised version)
* * *
**Comment #16**

*L. 553 "F. albida" Italics is needed here.*

**Response #16**

Corrected

Line 594 (Revised version)
* * *
**Comment #17**

*L. 617 "roots" Root*

**Response #17**

Corrected

Line 658 (Revised version)
* * *
**Comment #18**

*L. 634 I suggest deleting these two abbreviations (AF and FS).*

**Response #18**

The deletion was made

Line 675 to 683 (Revised version)
* * *
**Comment #19**

*L. 683 "have been also" Have also been.*

**Response #19**

Corrected

Line 725 (Revised version)
* * *
**Comment #20**

*L. 692 "and cowpeas" I do not understand why this is reported here since this crop was not grown during the experimental period.*

**Response #20**

The field where the EC flux tower is installed features peanut cultivation intercropped with cowpea. Therefore, the ecosystem respiration fluxes (Reco.EC) measured by the tower include the contribution of cowpea to respiration.
* * *
**Comment #21**

*L. 696 "field's" Field.*

**Response #21**

Corrected

Line 738 (Revised version)
* * *
**Comment #22**

*L. 710 "footprint's" Footprint.*

**Response #22**

Corrected

Line 752 (Revised version)
* * *
**Comment #23**

*L. 727 "compartment's" Compartment.*

**Response #23**

Corrected

Line 762 (Revised version)
* * *
**Comment #24**

*L. 734 "advancing understanding" Advancing the understanding.*

**Response #24**

Corrected

Line 770 (Revised version)
* * *
**Comment #25**

*L. 740, 767, 772 "system's" System.*

**Response #25**

Corrected

Lines 781, 811 and 816 (Revised version)
* * *
**Comment #26**

*L. 770, 794, References "F. albida" Italics is needed here.*

**Response #26**

Corrected

Lines 816 and 847 (Revised version)

---

## Author Comment (AC3)

**Title:** *Drivers and CO₂ flux budgets in a Sahelian Faidherbia albida agro-silvo-pastoral parkland: Insights from continuous high-frequency soil chamber measurements and Eddy Covariance.*

**Author (s):** Seydina Mohamad Ba et al.

**MS type:** Original research article

**Manuscript No.:** egusphere-2025-2660, submitted to Soil
* * *
Dear Dr Jim Boonman,

We were delighted to receive your detailed and highly constructive comments on our manuscript entitled "Drivers and $CO_2$ flux budgets in a Sahelian Faidherbia albida agro-silvo-pastoral parkland: Insights from continuous high-frequency soil chamber measurements and eddy covariance".

We sincerely thank you for agreeing to review our work and for the considerable time and effort you have invested in it. We have carefully considered every one of your suggestions and have implemented the recommended changes, which have substantially improved the clarity and overall quality of the paper.

Please note that the line numbers mentioned in the responses are provided on a provisional basis, as further corrections and revisions of the manuscript at the next step may result in changes to the line numbering. Consequently, this will be taken into account in the final revised version.

Please find below our point-by-point responses to your comments.

With best regards,

On behalf of all the co-authors,

**Seydina Mohamad BA,**
PhD candidate in the EU-funded CASSECS project
IESOL, Centre IRD-ISRA, 18524, Dakar; Senegal
Email: seydina.ba@ird.fr
and
**Olivier Roupsard,**
Researcher
CIRAD, UMR Eco&Sols, Dakar; Sénégal
Eco&Sols, Univ Montpellier, CIRAD, INRAE, Institut Agro, IRD, Montpellier; France
IESOL, Centre IRD-ISRA de Bel Air, 18524, Dakar; Senegal
Email: olivier.roupsard@cirad.fr

Response guidelines:

The comments from Reviewer RC2 are reproduced in *black and in italic*. The authors'
responses are provided in black and in regular (non-italic) text. The revised text or lines
added/changed lines from the manuscript are shown in blue and in bold.
* * *
**Authors' response to reviewer RC2**

**Reviewer's report:**

*The research of Ba et al. focuses on $CO_2$ flux dynamics of an increasingly popular agro-forestry land use in Africa, featuring Faidherbia albida trees, groundnut plants and livestock grazing. All chapters are neatly written which gives the reader a full and clear picture of the research that has been done and the results that were collected. Overall, the research seems to be conducted well and features interesting findings. Most of all, the authors have quantified GPP and Rh of various ecosystem elements and showed how these elements fit within the bigger picture of the complete ecosystem. This increases our general understanding of these systems, which is needed to enable improvements of land-use in the longer term. Moreover, the results that are presented can be of high value for ecosystem and/or climate models as the (Sahel) region seems to be, as the authors state, particularly underrepresented in global carbon flux research. Nevertheless, I do have three major concerns or questions that I would like to mention below.*

**Comment #1**

*First, I am concerned about the methodology to partition and extrapolate CO2 fluxes discussed in section 2.3.3. The authors discuss the assumptions on which the Arrhenius-type function from Lloyd & Taylor (1994) relies that was used for extrapolating ecosystem respiration. The first assumption features an exponential response between soil temperature and respiration. However, the authors also describe that high (soil) temperatures suppress daily respiration (discussion section 4.1). This is attributed to a decreased microbial activity which suppressed soil respiration and has been described more often in literature. The authors not only found that respiration is suppressed at higher temperatures, but they also even mention a (weak) negative correlation between respiration and soil temperatures (section 3.4). Figure S2.6 shows, besides a suppression of Rch due to high temperatures, that there does not seem to be a clear exponential temperature relation. This raises questions about the validity of the assumption on which*

*the partitioning, extrapolation and gap-filling of CO2 fluxes were based. The authors show that the nocturnal respiration can be modelled quite well in Fig. 3, but how does this translate to daytime when the temperatures are higher? Given the observed negative correlation, the authors should justify their approach. If the model is inappropriate under high temperatures, a different approach might be needed.*

**Response #1**

The reviewer questions the validity of estimating daytime soil respiration from nighttime values modelled using the Lloyd and Taylor (1994) function. This is a highly pertinent and constructive comment.

In the present study, soil temperature ranges during day and night are remarkably similar. For instance, in full-sun conditions (away from trees), daytime soil temperature varies from a minimum of 20.7 °C (shortly after sunrise) to a maximum of 45.8 °C (around 17:00), whereas nighttime temperature reaches a minimum of 22.1 °C (just before dawn) and a near-maximum of ~45 °C immediately after sunset. Consequently, nocturnal and diurnal respiratory processes can reasonably be assumed to occur, a priori, under very similar thermal conditions. The model therefore remains applicable provided that the temperature range used for calibration encompasses the full thermal regime experienced over the diel cycle.

Furthermore, we emphasise that the Lloyd and Taylor (1994) model employed here is purely empirical, yet has become a widely adopted standard in the literature, making it particularly suitable for large-scale comparative studies and meta-analyses. We therefore consider the use of this model to be fully justified in the present context.

To address the reviewer's concern and to enhance clarity for readers, a concise introductory statement has been added at the start of Section 4 ("Discussion"):

"Prior to the discussion, it is important to recall the methodological basis underlying the modelling of soil respiration.

An empirical Arrhenius-type equation, as proposed by Lloyd and Taylor (1994) and widely adopted in the literature, was used to model soil respiration as a function of soil temperature. In the present study, daytime soil respiration was calculated from the modeled values of nighttime soil respiration, assuming comparable thermal conditions between day and night. Indeed, measured soil temperature is remarkably similar (20.7—

45.8°C during the day versus 22.1— 45.0°C at night; data not shown), thereby ensuring that the model calibration encompassed the full diurnal thermal cycle.

The model parameters were recalibrated every five days, which represents a methodological compromise between the temporal resolution and the robustness of data. This rolling calibration allowed us to capture the seasonal variability of soil respiration while maintaining sufficient stability for reliable model parameters estimation.

Overall, this approach provides a consistent and empirically grounded framework for estimating diel CO2 exchange dynamics within the system; data not shown), thereby ensuring that the model calibration encompassed the full diurnal thermal cycle.

The model parameters were recalibrated every five days, which represents a methodological trade-off between the temporal resolution and the robustness of data. This rolling calibration allowed us to capture the seasonal variability of soil respiration while maintaining sufficient stability for reliable model parameter estimation.

Overall, this approach provides a consistent and empirically grounded framework for estimating diel $CO_2$ exchange dynamics within the system".

Line 556 to 570 (Revised version)

**Comment #2**

*Second, I have a question about non-linearity which could affect chamber flux results. When working with the chambers the authors noted fogging and decided to shorten the flux-analysis from 15 to 5 minutes during the groundnut growing season. Other causes may still lead to a non-linear measurement of CO2 concentrations after chamber closure. For example, a high plant uptake of CO2 could diminish CO2 concentrations substantially, eventually slowing down plant uptake. When a flux is calculated using a fitting period that is too long, the slope of the CO2 uptake will be lower than the initial slope, misrepresenting the actual initial CO2 uptake and affecting the total CO2 balance. How did the authors make sure this non-linearity was minimized during the flux calculations? Did some 5-minute flux measurements turn out to be non-linear? If so, how were these cases handled? Were any non-linear fluxes excluded by filtering fluxes that had a R2 < 0.8? Would that be the right choice?*

**Response #2**

We thank the reviewer for this pertinent comment regarding the potential non-linearity of $CO_2$ concentration increase following chamber closure.

In the present study, we observed no substantial deviation from linearity, even when measurement duration was extended from 5 to 15 minutes. Moreover, the relatively large chamber headspace height (0.50 m) helps maintain a stable concentration gradient over time, in contrast to the much smaller chambers sometimes used in other studies. These design features ensure that measured fluxes remain linear throughout the entire recording period.

We are willing to provide, as an appendix if desired, the raw recordings of the temporal evolution of $CO_2$ in the chambers over several cycles, showing that the slope does not change when the chambers remain closed for extended periods (15 minutes).

Regarding data filtering, an $R^2 > 0.8$ threshold was applied to exclude measurements that could be compromised by water ingress into the tubing during the rainy season, by water-vapour condensation, or by any other artefacts likely to bias the flux estimates (e.g., incomplete chamber closure).
* * *
**Comment #3**

*Third, the authors present an annual carbon budget of the ecosystem that was measured but did not include harvest and livestock manure C-terms. Even though the authors clearly mention and discuss this problem in the methods and discussion sections, I have my doubts about the usage of the term carbon budget. When the livestock was not fed externally, and manure is not exported from the system, we could assume that the presence and grazing would have a marginal impact on the carbon budget. However, in the discussion it is mentioned that faeces are collected from the field. Furthermore, biomass harvest C-export normally represents a substantial term within a carbon budget of an agricultural system. I do understand that a carbon budget is a valuable result. However, ignoring these C-terms and then comparing the carbon budget to literature seems incorrect and may lead to misleading comparisons. Would it be possible to roughly estimate the missing components to construct an actual carbon budget? The estimates could feature substantial errors that can be propagated. Such an approach may provide a more complete carbon budget and facilitate a fair comparison with other studies.*

**Response #3**

We thank the reviewer for this highly pertinent comment. We acknowledge that neither carbon exports associated with biomass harvesting nor carbon inputs from animal excreta were quantified in the present study. We agree that the use of the term "carbon budget" may be misleading, and we have therefore clarified at line 297 to 299 (revised version) that the budget calculated here is only apparent. This means that the budget calculated here represents only the balance of vertical CO2 fluxes between the soil, the vegetation and the atmosphere, excluding lateral C fluxes such as biomass export/import and free manure return from animals.

Our objective was to provide a first integrated estimate of the major vertical $CO_2$ fluxes (photosynthesis, respiration, and net ecosystem exchange) based on two complementary approaches (chamber-based vs. eddy-covariance), rather than to deliver a complete carbon budget. This is now explicitly stated at lines 301–303 (revised version). Accordingly, throughout the revised version, we have systematically replaced "annual C budget" with "annual vertical $CO_2$ balance".

Lines 1, 52, 293, 297, 549, 555; 798 and 828 (Revised version)

We have also added the following to Section 4.7 (Limitations of the Study) for further clarification:

"Furthermore, the present study constitutes only an intermediate step delivering a first integrated estimate of the main vertical $CO_2$ exchanges (photosynthesis, respiration, and net ecosystem exchange) as a base for a forthcoming paper that will present a more comprehensive carbon budget of the ecosystem. Establishing such a carbon budget would require substantial additional data acquisition and poses considerable methodological challenges. In particular, quantifying carbon inputs/outputs associated with free-ranging livestock grazing would be difficult to achieve with acceptable accuracy. It must also be recognised that the system is in a dynamic, non-steady state, characterised by marked inter-annual variability as well as periods of carbon storage and release, which are difficult to constrain empirically except through modeling".

Line 827 to 835 (Revised version).

As a reminder, the study site has been equipped since 2018 with an eddy-covariance flux tower installed above the tree canopy, providing a continuous multi-year time series of ecosystem-scale $CO_2$ exchange. However, as is also the case for other carbon-budget

studies conducted in the Sahel (Tagesson et al., 2015; Wieckowski et al., 2024), these data represent the balance of vertical $CO_2$ fluxes only.

In parallel, several complementary agronomic studies, particularly those quantifying harvested biomass, are currently underway. These ongoing efforts will enable us, in a forthcoming and more comprehensive article, to substantially refine the budget presented here and to construct a more realistic and complete carbon budget.

**Comment #4**

*The highlights include abbreviations (Sh, FS) that are unknown to readers.*

**Response #4**

The necessary clarifications have been added to ensure that these acronyms are readily understandable to readers.

Line 34 to 35 (Revised version)

**Comment #5**

*Line 99. Please check the usage of present time.*

**Response #5**

The use of the present indicative has been reviewed, and the sentence has been rephrased as follows: "Specifically, the study aims to (1) conduct year-round, high-frequency in situ $CO_2$ flux measurements from soil and crops using automated static chambers; (2) partition the net $CO_2$ fluxes (FCO$_2$ch) into respiration (Rch) and photosynthesis (GPPch); (3) investigate the environmental drivers of fluxes and the spatial variability linked to tree presence; and (4) compare chamber-based flux estimates with ecosystem-scale measurements derived from the EC method".

Line 102 to 107 (Revised version)

**Comment #6**

*Line 225. Please remove the repetition.*

**Response #6**

Repetition has been removed.

Line 231 to 234 (Revised version)

**Comment #7**

*Line 443. Table 1 results for daily FCO2 are negative, while numbers here appear positive.*

**Response #7**

We thank the reviewer for this comment. Indeed, in the table, $FCO_2$ch values (annual sum and mean values) are reported as negative (Line 481, revised version). However, when comparing in mean magnitudes between full-sun (FS) and shaded (Sh) conditions in the main text, $FCO_2$ch was expressed as an absolute value.
To eliminate this potential source of confusion, we have explicitly stated throughout the manuscript, where relevant, that $FCO_2$ch values are reported "in absolute terms" when presented as mean values in the main text.

Lines 466 and 468 (Revised version)

**Comment #8**

*Table 2. It is a choice to not denote non-significant correlations. However, a p-value of 0.05 is arbitrary. There might be different visions on this matter, but I would not 'hide' non-significant correlations and show each p-value (or p-value category).*

**Response #8**

The p-values have been added to Table 2.
Line 499 (revised version)

**Comment #9**

*Table 4. How was the std error that is shown calculated?*

**Response #9**

We thank the reviewer for pointing this out. An error had indeed been made in the uncertainty estimation. The reported standard error was initially based on the daily mean standard deviation, implicitly assuming that the uncertainty remained constant regardless of the number of measurement days. This approach is only valid when calculating the uncertainty of an annual mean flux, not when estimating the uncertainty of an annual cumulative flux.

We have therefore corrected this and added the appropriate clarification at the end of Section 2.4 ("Statistical analyses").

"The standard error of the total annual flux was estimated using the error propagation method. This calculation considered the mean standard deviation of daily fluxes (g C–CO$_2$ d$^{-1}$) and the effective number of measurement days (365). For each FS and Sh condition, the mean daily standard deviation was multiplied by the square root of 365 to obtain the annual standard error. The resulting values were then weighted by 90% for FS and 10% for Sh to derive the overall standard error of the annual flux sum, which was subsequently converted to Mg C–CO$_2$ ha$^{-1}$".

Line 345 to 350 (Revised version)

The necessary corrections for standard error values have also been applied throughout the entire manuscript.

Lines 52, 53, 551 to 556, 559, 779 to 780, 786, 804, (Revised version)
* * *
**Comment #10**

*Section 4.5. Sometimes it is hard to follow which periods are being discussed. In general, it could help to specifically mention the months that are being discussed.*

*Line 704. The authors mention that chamber and EC GPP measurements agree closely. I do agree that this is the case in August, but after the beginning of September the two seem to start deviating remarkably. As mentioned above, please clarify which months are under discussion.*

**Response #10**

When referring to the agreement between chamber-derived GPP and eddy-covariance (EC) estimates, we meant concordance at two distinct levels: 1) temporal dynamics (restricted to the rainy season), which exhibit highly similar patterns between the two methods until peanut harvest in the chambers, and 2) flux magnitude, with particularly strong agreement during the month of August, as the reviewer rightly highlighted.

Accordingly, we have added the corresponding clarification in Section 4.

Line 734 to 736 (Revised version)
* * *
**Comment #11**

*Section 4.6. Please see the third point above.*

**Response #11**

Checked.
* * *
**Comment #12**

*Line 794. Since the actual carbon balance is unknown, it cannot be stated that the agroforestry systems that were studied are 'effective carbon sinks'.*

**Response #12**

We have added the necessary clarifications in this regard. The sentence now reads:

« Sustainable management practices, particularly regarding C inputs/outputs from the system regarding crop harvest, residues exporting, and cattle free manuring must be taken into account to confirm the system capacity to act as a carbon sink".

Line 809 to 812 (Revised version)

---

## Author Comment (AC4)

**Title:** *Drivers and* **vertical** *$CO_2$ flux* **balances** *in a Sahelian Faidherbia albida agro-silvo-pastoral parkland: Insights from continuous high-frequency soil chamber measurements and Eddy Covariance.*

**Author (s):** Seydina Mohamad Ba et al.

**MS type:** Original research article

**Manuscript No.:** egusphere-2025-2660, submitted to Soil
* * *
Dear Editor and Riccardo Picone,

We would like to thank you once again for taking the time to carefully examine the authors' responses to the various questions and comments you had previously raised. We then received a series of minor revisions from you, which we have fully taken into account and incorporated into the revised version of the manuscript. We greatly appreciate this feedback, as it has significantly improved the quality of our article.

Please find below the additional revisions requested by the editor and an updated version of the authors' responses to the reviewer RC1. For clarity:

- ▪ Editor comments are shown in *red italics*;
- ▪ Additional revisions are presented in **green bold text**;
- ▪ The comments of reviewer are presented in *black italics*;
- ▪ Updated responses, with final line numbers, appear in plain black text, and changes made to the manuscript are indicated in **blue bold**.

With best regards,

On behalf of all the co-authors,

Seydina Mohamad BA,
PhD candidate in the EU-funded CASSECS project
IESOL, Centre IRD-ISRA, 18524, Dakar; Senegal
Email: seydina.ba@ird.fr
and
Olivier Roupsard
Researcher
CIRAD, UMR Eco&Sols, Dakar; Sénégal
Eco&Sols, Univ Montpellier, CIRAD, INRAE, Institut Agro, IRD, Montpellier; France
IESOL, Centre IRD-ISRA de Bel Air, 18524, Dakar; Senegal
Email: olivier.roupsard@cirad.fr

**Additional revisions requested by the editor**

**Authors' responses to the editor**

Editor's report:

Comment #7

*The response #7 on comment #7 by reviewer 1 (Riccardo Picone) is not yet entirely clear. A more detailed explanation of a measurement sequence, including the measurement frequency, chamber closure time and purging time, could contribute to the clarity of the manuscript.*

Response #7

The main automated steps of the $CO_2$ flux measurement cycle (purging, chamber closure, measurement delay, $CO_2$ accumulation measurements, and chamber reopening) have been provided in Supplementary Material S1, in Table S1.2.

**Updated authors' responses to the reviewer RC1**

Reviewer's report:

*This work reports a comparison of carbon fluxes between different zones of an agroforestry system during one whole year assessed with two different methodologies. A comparison of the two methodologies was also done. The topic is therefore highly relevant for the journal. The abstract does a good work in framing the context of the study and the relevance of the findings. The introduction effectively presents the topic and its importance, by highlighting key knowledge gaps that will be addressed by the study. Anyway, I would suggest including a brief description of the Eddy Covariance method in this section. In my opinion, the main problem of the paper is that clear starting hypotheses that have driven the work were not stated. This should be addressed. Methodologies appear to be consistent and appropriately described, and all the reported methods have the appropriate bibliographic reference. I would only suggest some minor integrations to the experimental design description. Results are correctly reported in all the necessary detail. The discussion does a very good job in comparing the results to other studies, hypothesizing mechanisms driving the findings, and highlighting limitations of the study. Conclusions realistically summarize the key discoveries. All supplementary materials are relevant and correctly reported. The authors are requested to carefully proofread the "references" section because some journal names are not correctly abbreviated. Based on these considerations, I would recommend minor revisions to be applied to the manuscript before it can be accepted for publication.*

*Hereafter follow the specific comments I made on the text. Text between quotation marks indicates citations from the manuscript. When multiple lines are indicated, the comments refer either to a full sentence or to a meaningful part of it*

—--------------------------------------------------------------------------------------------------------------

**Comment #1**

*L. 43 Please report the full name of the species when it is first mentioned in the abstract.*

**Response #1**

The full name of the species was mentioned

Line 43: "*Faidherbia albida*" (Revised version)

**Comment #2**

L. 57 I would suggest deleting the phrase "of trees".

**Response #2**

The correction was done.

Line 57: "fertile island effect" (Revised version)
* * *
**Comment #3**

*L. 64 I would suggest briefly describing the methodology used for this technique.*

**Response #3**

The methodology used for EC technique is described as follows:

"The EC technique quantifies $CO_2$ exchanges between ecosystems and the atmosphere by correlating fluctuations in vertical wind velocity with simultaneous variations in $CO_2$ concentrations, providing a direct and non-invasive estimate of $CO_2$ fluxes (Baldocchi, 2003)."

Line 66 to 69 (Revised version)
* * *
**Comment #4**

*L. 95 "upscaling" Undertsanding/comprehension?*

**Response #4**

Thank you for pointing this out.

The main idea, here, is to partition the ecosystem fluxes by compartments (soil and trees).

The corrected sentence becomes:

"When combined with EC, this dual-method approach strengthens source attribution and improves the partitioning of fluxes across complex agroforestry landscapes".

Line 97 to 99 (Revised version)
* * *
**Comment #5**

*L. 99 The hypotheses that have driven the study are not stated.*

**Response #5**

Thank you very much for this comment, which further improves the quality of the paper.

We have therefore added the following assumptions:

(1) Rch and GPPch are higher under the canopy of F. albida than in full sun, (2) soil moisture is the main environmental factor directly controlling both Rch and GPPch, (3) when extrapolated to the field scale, the chamber-based method provides seasonal dynamics of respiration and photosynthesis fluxes comparable to those derived from EC technique.

Line 108 to 112 (Revised version)
* * *
**Comment #6**

*L. 139 At which distance from the trees were the chambers installed?*

**Response #6**

This was already mentioned in non-revised version line 139 (...at least 20 m from trees).

Now in line 148 (Revised version)
* * *
**Comment #7**

*L. 147 "half-hour flux measurements" Does this mean that the measurement was repeated every 30 mins in each chamber? If so, I suggest being more clear on this.*

**Response #7**

This question is now fully addressed by providing and supplement table in Supplementary Materials S1, in Table S1.2.
* * *
**Comment #8**

*L. 160 "indicated" Indicating?*

**Response #8**

The correction has been made.

Line 169: "indicating" (Revised version)
* * *
**Comment #9**

*L. 166, 168 "NDVI", "LAI" Please report the full name.*

**Response #9**

The full name for NDVI (normalised difference vegetation index) and LAI (leaf area index) was reported.

Lines 175 and 178 (Revised version)
* * *
**Comment #10**

*L. 172-174 How often were VWC and Tsoil measurements repeated?*

**Response #10**

"All at 5-min intervals", this measurement frequency for VWC and Tsoil was already indicated in non-revised version line 174.

Line 184 (Revised version).

**Comment #11**

*L.223-227 These two sentences are a repetition.*

**Response #11**

The repetition has been removed.

Line 230 to 232 (Revised version)

**Comment #12**

*L. 344 Inside "the" chambers.*

**Response #12**

The missing article ''the" has been added.

Line 354 (Revised version)

**Comment #13**

*L.351 "references" Reference*

**Response #13**

The correction has been made.

Line 362 (revised version)

**Comment #14**

*L.s 387-388 How do you account for a standard error of the same entity of the measurement itself?*

**Response #14**

Here, it refers to the mean $\pm$ standard deviation, not the standard error. I have added this clarification in the text.

The sentence now reads: "FS showing (mean $\pm$ standard deviation) $0.9 \pm 0.9$ µmol $CO_2$ $m^{-2}$ $s^{-1}$ (modeled) and $1.3 \pm 1.2$ µmol $CO_2$ $m^{-2}$ $s^{-1}$ (measured)".

Line 398 – 399 (Revised version).
* * *
**Comment #15**

*L. 483 "GPPshowed" A space is needed here.*

**Response #15**

Corrected.

Line 476 (Revised version)
* * *
**Comment #16**

*L. 553 "F. albida" Italics is needed here.*

**Response #16**

Corrected

Line 590 (Revised version)
* * *
**Comment #17**

*L. 617 "roots" Root*

**Response #17**

Corrected

Line 654 (Revised version)
* * *
**Comment #18**

*L. 634 I suggest deleting these two abbreviations (AF and FS).*

**Response #18**

The deletion was made

Line 669 to 678 (Revised version)
* * *
**Comment #19**

*L. 683 "have been also" Have also been.*

**Response #19**

Corrected

Line 721 (Revised version)
* * *
**Comment #20**

*L. 692 "and cowpeas" I do not understand why this is reported here since this crop was not grown during the experimental period.*

**Response #20**

The field where the EC flux tower is installed features peanut cultivation intercropped with cowpea. Therefore, the ecosystem respiration fluxes (Reco.EC) measured by the tower include the contribution of cowpea to respiration.

Lines 273 to 276 (Revised version)
* * *
**Comment #21**

*L. 696 "field's" Field.*

**Response #21**

Corrected

Line 734 (Revised version)
* * *
**Comment #22**

*L. 710 "footprint's" Footprint.*

**Response #22**

Corrected

Line 748 (Revised version)
* * *
**Comment #23**

*L. 727 "compartment's" Compartment.*

**Response #23**

Corrected

Line 766 (Revised version)
* * *
**Comment #24**

*L. 734 "advancing understanding" Advancing the understanding.*

**Response #24**

Corrected

Line 773 (Revised version)
* * *
**Comment #25**

*L. 740, 767, 772 "system's" System.*

**Response #25**

Corrected

Lines 777, 806 and 812 (Revised version)
* * *
**Comment #26**

*L. 770, 794, References "F. albida" Italics is needed here.*

**Response #26**

Corrected

Lines 810 and 843 (Revised version)

---

## Author Comment (AC5)

**Title:** *Drivers and **vertical** CO₂ flux **balances** in a Sahelian Faidherbia albida agro-silvo-pastoral parkland: Insights from continuous high-frequency soil chamber measurements and Eddy Covariance.*

**Author (s)**: Seydina Mohamad Ba et al.

**MS type**: Original research article

**Manuscript No.:** egusphere-2025-2660, submitted to Soil
* * *
Dear Editor and Dr Jim Boonman,

We sincerely thank you for carefully reviewing our initial responses and for providing additional suggestions that further strengthen the scientific rigor and quality of the manuscript.

One of the main points raised by the editor concerns the validity of using the Arrhenius equation to model respiration fluxes in the Sahelian context. As rightly pointed out by the editor, Arrhenius-type equations assume an exponential dependence on temperature and generally lose validity beyond a certain thermal threshold. This limitation is particularly critical in our study area, where temperatures frequently exceed 40°C. We fully acknowledge this concern; however, we would like to clarify that the model used in our study is not based on the classical Arrhenius formulation, but on a modified version proposed by Lloyd and Taylor (1994) (see **Response #1**). Nevertheless, based on a detailed analysis of the available data, we re-evaluated the model to assess its validity at high temperatures, discussing its limitations under extreme conditions and exploring alternative approaches. Minor revisions recommended by the editor have also been fully addressed, and the responses to the reviewer have consequently been updated.

Please find below the authors' responses to your report, including all additional revisions requested by the editor, as well as an updated version of the previous responses to the reviewer RC2. For ease of reading:

- Editor comments are shown in *red italics*;
- Additional revisions are presented in **green bold text**;
- The comments of reviewer are presented in *black italics*;
- Updated responses, with final line numbers, appear in plain black text, and changes made to the manuscript are indicated in **blue bold**.

With best regards,

On behalf of all the co-authors,

Seydina Mohamad BA,
PhD candidate in the EU-funded CASSECS project
IESOL, Centre IRD-ISRA, 18524, Dakar; Senegal
Email: seydina.ba@ird.fr
and
Olivier Roupsard
Researcher
CIRAD, UMR Eco&Sols, Dakar; Sénégal
Eco&Sols, Univ Montpellier, CIRAD, INRAE, Institut Agro, IRD, Montpellier; France
IESOL, Centre IRD-ISRA de Bel Air, 18524, Dakar; Senegal
Email: olivier.roupsard@cirad.fr

**Additional revisions requested by the editor**

**Authors' responses to the editor**

Editor's report:

**Comment #1**

*The use of nighttime respiration data to calculate daytime respiration seems valid if thermal ranges observed during the night resemble the daytime temperature range. Furthermore, the rolling parameter recalibration methodology seems to be a good compromise. However, the discussion in the rebuttal seems to ignore the doubts raised by the reviewer about the use of Lloyd and Taylor's Arrhenius model. This model assumes an exponential response between soil temperature and respiration, whereas the authors later observe a decrease in respiration at high temperatures. It is widely known that the Arrhenius type of equations does not work for biological systems above a certain temperature. Generally, in greenhouse gas emission studies soil temperatures stay below the threshold above which modifications are needed to describe changes in enzyme kinetics. In some biological studies modified Arrhenius models are used or quadratic or double exponential models. As temperatures in the Sahel are often in this high temperature range it is worth exploring the use of an adjusted Arrhenius type model for $CO_2$ extrapolation.*

**Response #1**

We thank the editor for this insightful comment. We agree that the classical Arrhenius equation, when applied to soil respiration, can lead to overestimations of fluxes when temperatures exceed the optimal physiological range of biological systems. However, it is crucial to emphasize that the Lloyd and Taylor (1994) model used in this study represent a modified form of the classical Arrhenius law. Unlike the classical Arrhenius formulation, Lloyd and Taylor (1994) introduced the (Tsoil – T0) term in the denominator of the exponential expression (**see Eq. 4, line 237 in revised version**), which induces a progressive decline in the temperature sensitivity of soil respiration as temperature rise above a given threshold. This structural feature results in a flattening of the respiration–temperature relationship at high temperatures, thereby preventing the overestimations commonly observed with models based strictly on the classical Arrhenius law.

To explicitly address the concern raised by the editor and to assess the validity of the Lloyd and Taylor (1994) model beyond the identified temperature thresholds (32 °C for FS and 29.5 °C for Sh; **see Supplement Materials S2, Fig. S2.7 in updated version**), we conducted the following analyses:

**1. Residual analysis**

We analysed the residuals from the Lloyd and Taylor (1994) model to determine whether a systematic overestimation of nighttime respiration occurs when temperatures exceed the thresholds.

The results indicate that the mean of the residuals of the model, plotted as a function of soil temperature, remains generally centered around zero across the entire thermal gradient, including beyond the temperature thresholds (see Fig. S2.4 below). No systematic pattern indicative of flux overestimation at high temperature was observed.

[Figure]

Fig. S2.4: Residuals of the Lloyd and Taylor (1994) model in (a) full sun and (b) away from trees. The blue curve represents the mean trend of the residuals as a function of soil temperature (Tsoil), and the light orange shading corresponds to the 95% confidence interval.

**2. Incorporating a thermal inhibition factor**

We tested the modified Lloyd and Taylor (1994) model that incorporates a thermal inhibition factor, inspired by the CENTURY model (Parton et al., 1987), and later adapted for agroecosystems by Delogu (2013) in her doctoral thesis. The objective was to determine whether adding an inhibition term above specific temperature thresholds

improves significantly model performance. This analysis was applied only to respiration fluxes measured outside the canopy (FS), in order to evaluate the soil thermal response under homogeneous microclimatic conditions directly representative of the measured soil temperature. This approach was chosen to isolate the effect of tree cover, given that it represents only about 10% of the total surface. In other words, if no thermal inhibition is observed for soil respiration in FS (which is significantly warmer), it can be confidently assumed that the same would apply under the shaded conditions (Sh).

$$f(T) = (\frac{45 - Tsoil}{45 - 20})^{2.5} * exp\left[1.25 * (1 - (\frac{45 - Tsoil}{45 - 20})^{2.5})\right] \text{ when Tsoil} > 32°C$$

The results show that the thermal inhibition factor slightly reduces model performance ($R^2 = 0.8$; RMSE = 0.5 µmol $CO_2$ $m^{-2}$ $s^{-1}$) compared to the Lloyd and Taylor formulation without a thermal inhibition factor ($R^2 = 0.9$; RMSE = 0.4 µmol $CO_2$ $m^{-2}$ $s^{-1}$) (see the figure below). This slight decline reflects an over-correction of temperature sensitivity, which is inherently accounted for in the structure of the Lloyd and Taylor (1994) model.

[Figure]

Fig: Comparison between the Lloyd and Taylor (1994) model and a modified version including an additional temperature inhibition factor (not shown in supplementary materials). Y axis represents modeled nocturnal respiration and X axis measured respiration.

PARTON, W. J., SCHIMEL, D. S., COLE, C. V. & OJIMA, D. S.: Analysis of factors controlling soil organic matter levels in Great Plains grasslands, Soil Sci. Soc. Am. J., 51, 1173–1179, https://doi.org/10.2136/sssaj1987.03615995005100050015x, 1987.

Elodie Delogu. Modélisation de la respiration du sol dans les agroécosystèmes. Sciences de la Terre. Université Paul Sabatier- Toulouse III, 2013. Français. NNT: tel-00953712, https://theses.hal.science/tel-00953712v1/file/DELOGU_ThA_se.pdf

In conclusion, these analyses collectively support the validity of the Lloyd and Taylor (1994) model used in this study. The results indicate that the model does not overestimate soil respiration at high temperatures and provides a realistic representation of the thermal dynamics of soil respiration without requiring an extra thermal inhibition factor. The model effectively captures the attenuation phase of soil respiration at elevated temperatures, which primarily reflects environmental constraints characteristic of semi-arid ecosystems. These constraints are more pronounced during the dry season than during the rainy season and mainly correspond to reductions in soil moisture, which decrease microbial activity, as well as, to a lesser extent, thermal stress affecting microbial enzymatic processes.

In the updated version of the supplementary materials, we have added Fig. S2.4, presenting the model residuals, which we consider sufficient to address the editor's concern. Consequently, the analysis regarding the incorporation of a thermal inhibition factor into the model has not been included in the Supplementary Materials.

Furthermore, we deemed it necessary to add a new section in ''Discussion 4.1'' in the revised manuscript, titled 'Soil respiration modelling and limitations regarding temperature''.

**4.1. Soil respiration modelling and limitations regarding temperature**

In this study the Lloyd and Taylor (1994) model, based on a modified Arrhenius-type formulation, was used to model nocturnal soil respiration fluxes for estimating daytime respiration. Unlike the classical Arrhenius equation, this model includes the ($T_{soil}$ – $T_0$) term in the denominator of the exponential expression (Eq. 4), which inherently limits the effects of high temperatures by progressively reducing the temperature sensitivity of soil respiration as temperatures rise above a given threshold. This structural feature produces a flattening of the respiration–temperature relationship at elevated temperatures, thereby preventing the overestimations (Lloyd and Taylor, 1994).

The Lloyd and Taylor model has successfully been widely applied, primarily in boreal and temperate ecosystems (Lasslop et al., 2010; Reichstein et al., 2003), and relies on the assumption of comparable thermal conditions between daytime and nighttime periods (Juszczak et al., 2012). In our study, instantaneous soil temperatures ranged from 20.7 to 45.8 °C during the day and from 22.1 to 45.0 °C at night, indicating largely overlapping thermal ranges between the two periods. Model parameters were recalibrated using five-day fixed windows, which provided sufficient temporal resolution while capturing seasonal dynamics of soil respiration.

This study represents one of the first applications of the Lloyd and Taylor model in a Sahelian semi-arid context. While Arrhenius based models are known to potentially overestimate fluxes under extreme temperatures due to physiological limitations (Somero, 2020), over the range of temperatures observed in this study, the modeled soil respiration was not overestimated (Fig. S2.4). Thus, the model used in this study appears to provide a realistic representation of soil respiration under local conditions. However, this conclusion is site-specific and should not be interpreted as a general validation of temperature-based models across all semi-arid environments. Such models should be systematically validated with respect to temperature to ensure their reliability.

Line 556 to 579 (Revised version)
* * *
**Comment #2**

*The concern #2 by Jim Boonman are addressed adequately but will be even stronger when R2 0.8 threshold is incorporated in the material and a selection of raw recordings of temporal $CO_2$ measurements in the chambers will be shared in the appendix.*

**Response #2**

The $R^2 \geq 0.8$ threshold was incorporated (**line 202 in the revised version**), and a selection of raw $CO_2$ recordings from the chambers was added in the updated Supplementary Materials S1 (**Fig. S1.2**).

[Figure]

Fig. S1.2: $CO_2$ concentration time

**Comment #3**

*The concern #3 is solved by the adjustments (annual vertical CO2 balance) and it is clear that there are currently not enough data to provide a full carbon budget. When comparing the data to literature in the discussion it is important to explicitly mention which studies are fully comparable (those that focus on the vertical CO2 balance) and which studies include carbon import and export in their C balance.*

**Response #3**

The necessary clarifications have been provided in Section 4.7, specifying only studies reported vertical $CO_2$ fluxes and those accounted for horizontal fluxes associated with organic matter inputs or exports (lines 775 to 798).

**Updated authors' responses to the reviewer RC2**

Reviewer's report:

*The research of Ba et al. focuses on CO2 flux dynamics of an increasingly popular agroforestry land use in Africa, featuring Faidherbia albida trees, groundnut plants and livestock grazing. All chapters are neatly written which gives the reader a full and clear picture of the research that has been done and the results that were collected. Overall, the research seems to be conducted well and features interesting findings. Most of all, the authors have quantified GPP and Rh of various ecosystem elements and showed how these elements fit within the bigger picture of the complete ecosystem. This increases our general understanding of these systems, which is needed to enable improvements of land-use in the longer term. Moreover, the results that are presented can be of high value for ecosystem and/or climate models as the (Sahel) region seems to be, as the authors state, particularly underrepresented in global carbon flux research. Nevertheless, I do have three major concerns or questions that I would like to mention below.*

**Comment #1**

*First, I am concerned about the methodology to partition and extrapolate CO2 fluxes discussed in section 2.3.3. The authors discuss the assumptions on which the Arrhenius-type function from Lloyd & Taylor (1994) relies that was used for extrapolating ecosystem respiration. The first assumption features an exponential response between soil temperature and respiration. However, the authors also describe that high (soil) temperatures suppress daily respiration (discussion section 4.1). This is attributed to a decreased microbial activity which suppressed soil respiration and has been described more often in literature. The authors not only found that respiration is suppressed at higher temperatures, but they also even mention a (weak) negative correlation between respiration and soil temperatures (section 3.4). Figure S2.6 shows, besides a suppression of Rch due to high temperatures, that there does not seem to be a clear exponential temperature relation. This raises questions about the validity of the assumption on which the partitioning, extrapolation and gap-filling of CO2 fluxes were based. The authors show that the nocturnal respiration can be modelled quite well in Fig. 3, but how does this translate to daytime when the temperatures are higher? Given the observed negative*

*correlation, the authors should justify their approach. If the model is inappropriate under high temperatures, a different approach might be needed.*

**Response #1**

Responding and updating in additional revisions requested by the editor

**Comment #2**

*Second, I have a question about non-linearity which could affect chamber flux results. When working with the chambers the authors noted fogging and decided to shorten the flux-analysis from 15 to 5 minutes during the groundnut growing season. Other causes may still lead to a non-linear measurement of CO2 concentrations after chamber closure. For example, a high plant uptake of CO2 could diminish CO2 concentrations substantially, eventually slowing down plant uptake. When a flux is calculated using a fitting period that is too long, the slope of the CO2 uptake will be lower than the initial slope, misrepresenting the actual initial CO2 uptake and affecting the total CO2 balance. How did the authors make sure this non-linearity was minimized during the flux calculations? Did some 5-minute flux measurements turn out to be non-linear? If so, how were these cases handled? Were any non-linear fluxes excluded by filtering fluxes that had a R2 < 0.8? Would that be the right choice?*

**Response #2**

We thank the reviewer for this pertinent comment regarding the potential non-linearity of $CO_2$ concentration increase following chamber closure.

In the present study, we observed no substantial deviation from linearity, even when measurement duration was extended from 5 to 15 minutes. Moreover, the relatively large chamber headspace height (0.50 m) helps maintain a stable concentration gradient over time, in contrast to the much smaller chambers sometimes used in other studies. These design features ensure that measured fluxes remain linear throughout the entire recording period.

We are willing to provide, as an appendix if desired, the raw recordings of the temporal evolution of $CO_2$ in the chambers over several cycles, showing that the slope does not change when the chambers remain closed for extended periods (15 minutes).

Regarding data filtering, an $R^2 > 0.8$ threshold was applied to exclude measurements that could be compromised by water ingress into the tubing during the rainy season, by water-vapour condensation, or by any other artefacts likely to bias the flux estimates (e.g., incomplete chamber closure).
* * *
**Comment #3**

*Third, the authors present an annual carbon budget of the ecosystem that was measured but did not include harvest and livestock manure C-terms. Even though the authors clearly mention and discuss this problem in the methods and discussion sections, I have my doubts about the usage of the term carbon budget. When the livestock was not fed externally, and manure is not exported from the system, we could assume that the presence and grazing would have a marginal impact on the carbon budget. However, in the discussion it is mentioned that faeces are collected from the field. Furthermore, biomass harvest C-export normally represents a substantial term within a carbon budget of an agricultural system. I do understand that a carbon budget is a valuable result. However, ignoring these C-terms and then comparing the carbon budget to literature seems incorrect and may lead to misleading comparisons. Would it be possible to roughly estimate the missing components to construct an actual carbon budget? The estimates could feature substantial errors that can be propagated. Such an approach may provide a more complete carbon budget and facilitate a fair comparison with other studies.*

**Response #3**

We thank the reviewer for this highly pertinent comment. We acknowledge that neither carbon exports associated with biomass harvesting nor carbon inputs from animal excreta were quantified in the present study. We agree that the use of the term "carbon budget" may be misleading, and we have therefore clarified at line 296-297 (revised version) that the budget calculated here is only apparent. This means that the budget calculated here represents only the balance of vertical $CO_2$ fluxes between the soil, the vegetation and the atmosphere, excluding lateral C fluxes such as biomass export/import and free manure return from animals.

Our objective was to provide a first integrated estimate of the major vertical $CO_2$ fluxes (photosynthesis, respiration, and net ecosystem exchange) based on two complementary

approaches (chamber-based vs. eddy-covariance), rather than to deliver a complete carbon budget. This is now explicitly stated at lines 301–302 (revised version). Accordingly, throughout the revised version, we have systematically replaced "annual C budget" with "annual vertical $CO_2$ balance".

We have also added the following to Section 4.8 (Limitations of the Study) for further clarification:

"Furthermore, the present study constitutes only an intermediate step delivering a first integrated estimate of the main vertical $CO_2$ exchanges (photosynthesis, respiration, and net ecosystem exchange) as a base for a forthcoming paper that will present a more comprehensive carbon budget of the ecosystem. Establishing such a carbon budget would require substantial additional data acquisition and poses considerable methodological challenges. In particular, quantifying carbon inputs/outputs associated with free-ranging livestock grazing would be difficult to achieve with acceptable accuracy. It must also be recognised that the system is in a dynamic, non-steady state, characterised by marked inter-annual variability as well as periods of carbon storage and release, which are difficult to constrain empirically except through modeling".

Line 823 to 831 (Revised version).

As a reminder, the study site has been equipped since 2018 with an eddy-covariance flux tower installed above the tree canopy, providing a continuous multi-year time series of ecosystem-scale $CO_2$ exchange. However, as is also the case for other carbon-budget studies conducted in the Sahel (Tagesson et al., 2015; Wieckowski et al., 2024), these data represent the balance of vertical $CO_2$ fluxes only.

In parallel, several complementary agronomic studies, particularly those quantifying harvested biomass, are currently underway. These ongoing efforts will enable us, in a forthcoming and more comprehensive work to substantially refine the budget presented here and to construct a more realistic and complete carbon budget.
* * *
**Comment #4**

*The highlights include abbreviations (Sh, FS) that are unknown to readers.*

**Response #4**

The necessary clarifications have been added to ensure that these acronyms are readily understandable to readers. Line 34 to 35 (Revised version)
* * *
**Comment #5**

*Line 99. Please check the usage of present time.*

**Response #5**

The use of the present indicative has been reviewed, and the sentence has been rephrased as follows: "Specifically, the study aims to (1) conduct year-round, high-frequency in situ $CO_2$ flux measurements from soil and crops using automated static chambers; (2) partition the net $CO_2$ fluxes ($FCO_2ch$) into respiration (Rch) and photosynthesis (GPPch); (3) investigate the environmental drivers of fluxes and the spatial variability linked to tree presence; and (4) compare chamber-based flux estimates with ecosystem-scale measurements derived from the EC method".

Line 102 to 107 (Revised version)
* * *
**Comment #6**

*Line 225. Please remove the repetition.*

**Response #6**

Repetition has been removed.

Line 230 to 232 (Revised version)
* * *
**Comment #7**

*Line 443. Table 1 results for daily FCO2 are negative, while numbers here appear positive.*

**Response #7**

We thank the reviewer for this comment. Indeed, in the table 1, $FCO_2ch$ values (annual sum and mean values) are reported as negative (Line 467, revised version). However, when comparing in mean magnitudes between full-sun (FS) and shaded (Sh) conditions in the main text, $FCO_2ch$ was expressed as an absolute value.

To eliminate this potential source of confusion, we have explicitly stated throughout the manuscript, where relevant, that $FCO_2ch$ values are reported "in absolute terms" when presented as mean values in the main text.

Lines 452 and 455 (Revised version)

**Comment #8**

*Table 2. It is a choice to not denote non-significant correlations. However, a p-value of 0.05 is arbitrary. There might be different visions on this matter, but I would not 'hide' non-significant correlations and show each p-value (or p-value category).*

**Response #8**

The p-values have been added to Table 2.

Line 485 (revised version)

**Comment #9**

*Table 4. How was the std error that is shown calculated?*

**Response #9**

We thank the reviewer for pointing this out. An error had indeed been made in the uncertainty estimation. The reported standard error was initially based on the daily mean standard deviation, implicitly assuming that the uncertainty remained constant regardless of the number of measurement days. This approach is only valid when calculating the uncertainty of an annual mean flux, not when estimating the uncertainty of an annual cumulative flux.

We have therefore corrected this and added the appropriate clarification at the end of Section 2.4 ("Statistical analyses").

"The standard error of the total annual flux was estimated using the error propagation method. This calculation considered the mean standard deviation of daily fluxes (g C–$CO_2$ $d^{-1}$) and the effective number of measurement days (365). For each FS and Sh condition, the mean daily standard deviation was multiplied by the square root of 365 to obtain the annual standard error. The resulting values were then weighted by 90% for FS and 10% for Sh to derive the overall standard error of the annual flux sum, which was subsequently converted to Mg C–$CO_2$ $ha^{-1}$".

Line 331 to 336 (Revised version)

The necessary corrections for standard error values have also been applied throughout the entire manuscript.

**Comment #10**

*Section 4.5. Sometimes it is hard to follow which periods are being discussed. In general, it could help to specifically mention the months that are being discussed.*

*Line 704. The authors mention that chamber and EC GPP measurements agree closely. I do agree that this is the case in August, but after the beginning of September the two seem to start deviating remarkably. As mentioned above, please clarify which months are under discussion.*

**Response #10**

When referring to the agreement between chamber-derived GPP and eddy-covariance (EC) estimates, we meant concordance at two distinct levels: 1) temporal dynamics (restricted to the rainy season), which exhibit highly similar patterns between the two methods until peanut harvest in the chambers, and 2) flux magnitude, with particularly strong agreement during the month of August, as the reviewer rightly highlighted. Accordingly, we have added the corresponding clarification in Section 4.6.

"However, no divergence was observed in August, and the intensity of the peak of GPP in September was similar in both methods, but from the onset of groundnut senescence, when weeds became the dominant photosynthetic contributors".

Line 739 to 741 (Revised version)
* * *
**Comment #11**

*Section 4.6. Please see the third point above.*

**Response #11**

Checked.
* * *
**Comment #12**

*Line 794. Since the actual carbon balance is unknown, it cannot be stated that the agroforestry systems that were studied are 'effective carbon sinks'.*

**Response #12**

We have added the necessary clarifications in this regard. The sentence now reads:

« Sustainable management practices, particularly regarding C inputs/outputs from the system regarding crop harvest, residues exporting, and cattle free manuring must be taken into account to confirm the system capacity to act as a carbon sink".

Line 805 to 808 (Revised version).